# ADAM THROUGH A SECOND-ORDER LENS

## ABSTRACT

Research into optimisation for deep learning is characterised by a tension between the computational efficiency of first-order, gradient-based methods (such as SGD and Adam) and the theoretical efficiency of second-order, curvature-based methods (such as quasi-Newton methods and K-FAC). We seek to combine the benefits of both approaches into a single computationally-efficient algorithm. Noting that second-order methods often only function effectively with the addition of stabilising heuristics (such as Levenberg-Marquardt damping), we propose *AdamQLR*: an optimiser combining damping and learning rate selection techniques from K-FAC (Martens & Grosse, 2015) with the update directions proposed by Adam, inspired by considering Adam through a second-order lens. We evaluate AdamQLR on a range of regression and classification tasks at various scales, finding an *untuned* AdamQLR setting achieves comparable generalisation performance vs runtime to *tuned* benchmarks.

## 1 INTRODUCTION

At the heart of any machine learning model is an optimisation problem, and at the heart of any training procedure is an optimisation algorithm. Most frequently seen in the literature are *first-order* optimisers such as SGD, Adam (Kingma & Ba, 2015) and their variants, but exploratory studies have also been performed on *second-order* algorithms such as quasi-Newton methods and K-FAC (Martens & Grosse, 2015). Broadly speaking, second-order algorithms aim to secure more rapid convergence to an optimal value of the objective function by making more principled individual updates, which in turn are more computationally costly than those employed by first-order methods. Combined with a generally more complicated implementation, second-order methods have not yet proven preferable to first-order approaches for most practitioners (Anil et al., 2021).

In part, this is a stability issue — by virtue of taking larger individual steps, second-order optimisers carry an increased risk of significantly worsening the objective value if their approximate understanding of curvature in objective space is a poor representation of the true space. Most second-order approaches thus *depend* on additional heuristics (such as curvature damping) for their viability. Heuristics commonly seen in first-order methods, such as weight decay or momentum applied to SGD, improve an already effective optimiser; by contrast, second-order methods' heuristics are *essential* components, without which the optimiser will perform unstably or ineffectively. It is then natural to ask how much these heuristics are responsible for the documented benefits of second-order optimisers, and whether they might similarly improve first-order techniques.

In this paper, we propose a damped automatic learning rate strategy, derived by applying K-FAC's damping and learning rate selection techniques to Adam. The result is an efficient, scalable algorithm whose untuned form competes strongly with tuned commonly-used optimisers, demonstrating robustness to its few remaining hyperparameters. After a review of related work in Section 2, we present the development of our algorithm in Section 3. We then justify our claims by experiment in Section 4 before Section 5 concludes. Our main contributions are as follows:

- To our knowledge, we present the first use of damping and second-order approximations to select learning rates in Adam
- We propose a variation of damping based on Adam's internal curvature estimates which, when applied to Adam's update proposals, outperforms classical damping from e.g. K-FAC
- We show our untuned method competes with methods using tuned hyperparameters, exhibiting robustness to hyperparameters while saving substantial tuning cost

## 2 RELATED WORK

First-order methods form the bread and butter of modern machine learning, with SGD and Adam (Kingma & Ba, 2015) being most frequently seen. Adam belongs to a class of *adaptive* first-order methods, which apply some kind of normalisation transformation to the observed gradients; other examples include Adagrad (McMahan & Streeter, 2010; Duchi et al., 2011) and RMSprop (Tieleman & Hinton, 2012). Balles & Hennig (2018) demonstrate that Adam essentially scales gradient signs by their variance. Zhang et al. (2018) show that Adam can be seen as a form of natural gradient mean field variational inference, whose mode-fitting behaviour is known to underestimate variance, corresponding to overestimating curvature in an optimisation task (see e.g. Figure 1.3 in Turner & Sahani (2011)). Zhang et al. (2019) use a noisy quadratic model to argue for the benefits of exponential moving averages and other components found in Adam. These methods achieve computational efficiency by using diagonal approximations or heuristics to understand curvature in the space, so ignore useful information which we seek to incorporate.

Optimisers employing second-order derivative information are seen more often in the optimisation literature than in practical machine learning projects. The family of *quasi-Newton* methods (Nocedal & Wright, 2006) is inspired by the appearance of the Hessian matrix in a Taylor series truncated at quadratic order; this matrix characterises curvature in the model parameters. Martens (2010) use the Hessian-vector product trick (Pearlmutter, 1994) to work implicitly with the exact Hessian. Other work modifies the Hessian to avoid degeneracies — a particular concern in saddle point-dense high-dimensional spaces (Pascanu & Bengio, 2014; Dauphin et al., 2014). Although not explicitly using second derivatives, SHAMPOO (Gupta et al., 2018) learns a factorised set of preconditioned matrices. However, in non-convex, non-quadratic spaces like we consider, the unaltered Hessian may be badly misleading, leading to divergence of the training loss.

Where the system is viewed as a probabilistic model, an alternative curvature characterisation is the Fisher information matrix, which gives rise to the natural gradient family of methods (Amari, 1998). Unlike the Hessian, the Fisher matrix characterises curvature in KL-divergence space between the predicted and ground truth probability distributions. Factorized Natural Gradient (Grosse & Salakhudinov, 2015) approximates the Fisher using a Gaussian graphical model, while the Kronecker-Factored Approximate Curvature (K-FAC) method (Martens & Grosse (2015) after an idea by Heskes (2000)) imposes a block-diagonal approximation to the Fisher and represents each block by a Kronecker product. Extensions to K-FAC include EKFAC (George et al., 2018), which learns the approximate Fisher in an eigenvalue-aligned basis. K-BFGS (Goldfarb et al., 2020) applies a similar factorisation strategy to the Hessian matrix, retaining theoretical guarantees from the classical BFGS optimiser (Broyden, 1970; Fletcher, 1970; Goldfarb, 1970; Shanno, 1970). Although K-FAC can be applied in distributed settings, this is somewhat complex (Osawa et al., 2019), and the use of Fisher curvature requires new expressions to be calculated for each different network architecture block. We also find K-FAC to suffer a much greater overfitting risk than our approach, and we are able to compete with the relatively complex Fisher curvature using an appealingly simple method.

Another line of work aims to accelerate first-order methods by dynamically adapting the learning rate to match the local optimisation dynamics. Originally this was predominantly done by imposing fixed learning rate schedules (Darken & Moody, 1990; Li & Arora, 2019; Xu et al., 2019; Loshchilov & Hutter, 2017; Smith et al., 2018), but recent developments involve more dynamic adaptations by hypergradients (Franceschi et al., 2017; Micaelli & Storkey, 2020; Donini et al., 2020; Lorraine et al., 2020; Clarke et al., 2022), online Bayesian optimisation (Jin et al., 2023), or explicitly constructing an optimisation framework around the unique characteristics of deep neural networks (Bernstein et al., 2023). Zhang et al. (2019) and Kwatra et al. (2023) adopt a similar quadratic model methodology to our work, but the latter compute a finite-difference approximation to this model rather than using the exact curvature information as we do, and introduces additional hyperparameters controlling an exploration/exploitation trade-off. Niu et al. (2023) uses a parallel approach to ours to incorporate momentum into L-BFGS (Liu & Nocedal, 1989). These methods generally suffer an increased cost over simpler strategies, whether to discover a schedule, compute hypergradients or essentially perform inline hyperparameter optimisation, which in turn requires a substantial validation dataset to be held aside.

**Algorithm 1** Adam (Kingma & Ba, 2015)

$\mathbf{m}_0, \mathbf{v}_0 \leftarrow \mathbf{0}$
**for** $t = 1, 2, \cdots$ until $\boldsymbol{\theta}$ converged **do**
   $\mathbf{g}_t \leftarrow \nabla_{\boldsymbol{\theta}} f(\boldsymbol{\theta}_{t-1})$
   $\mathbf{m}_t \leftarrow \beta_1 \mathbf{m}_{t-1} + (1 - \beta_1)\mathbf{g}_t$
   $\mathbf{v}_t \leftarrow \beta_2 \mathbf{v}_{t-1} + (1 - \beta_2)(\mathbf{g}_t \odot \mathbf{g}_t)$
   $\widehat{\mathbf{m}}_t \leftarrow \frac{\mathbf{m}_t}{1 - \beta_1^t}$
   $\widehat{\mathbf{v}}_t \leftarrow \frac{\mathbf{v}_t}{1 - \beta_2^t}$
   $\mathbf{d}_t \leftarrow \frac{\widehat{\mathbf{m}}_t}{\sqrt{\widehat{\mathbf{v}}_t} + \epsilon}$

   $\boldsymbol{\theta}_t \leftarrow \boldsymbol{\theta}_{t-1} - \alpha \mathbf{d}_t$
**end for**

**Algorithm 2** AdamQLR

$\mathbf{m}_0, \mathbf{v}_0 \leftarrow \mathbf{0}$
**for** $t = 1, 2, \cdots$ until $\boldsymbol{\theta}$ converged **do**
   $\mathbf{g}_t \leftarrow \nabla_{\boldsymbol{\theta}} f(\boldsymbol{\theta}_{t-1})$
   $\mathbf{m}_t \leftarrow \beta_1 \mathbf{m}_{t-1} + (1 - \beta_1)\mathbf{g}_t$
   $\mathbf{v}_t \leftarrow \beta_2 \mathbf{v}_{t-1} + (1 - \beta_2)(\mathbf{g}_t \odot \mathbf{g}_t)$
   $\widehat{\mathbf{m}}_t \leftarrow \frac{\mathbf{m}_t}{1 - \beta_1^t}$
   $\widehat{\mathbf{v}}_t \leftarrow \frac{\mathbf{v}_t}{1 - \beta_2^t}$
   $\mathbf{d}_t \leftarrow \frac{\widehat{\mathbf{m}}_t}{\sqrt{\widehat{\mathbf{v}}_t} + \epsilon}$
   Update learning rate $\alpha$ according to (3)
   Update damping $\lambda$ according to (2)
   $\boldsymbol{\theta}_t \leftarrow \boldsymbol{\theta}_{t-1} - \alpha \mathbf{d}_t$
**end for**

## 3 ADAMQLR

We consider the minimisation of an arbitrary function $f(\boldsymbol{\theta})$, which for our purposes will be the loss function of some network parameterised by $\boldsymbol{\theta}$.

### 3.1 FIRST- AND SECOND-ORDER METHODS

Many optimisation algorithms in machine learning take the form $\boldsymbol{\theta}_t \leftarrow \boldsymbol{\theta}_{t-1} - \alpha \mathbf{u}(\mathbf{g}_t)$, where $\alpha$ is a learning rate and $\mathbf{u}$ some update function. This function $\mathbf{u}$ may depend on an internal state and the gradient $\mathbf{g}_t$, but not on any higher derivative. Adopting the ML convention, we call such algorithms *first-order* optimisers. By contrast, *second-order* optimisers take the form $\boldsymbol{\theta}_t \leftarrow \boldsymbol{\theta}_{t-1} - \mathbf{C}^{-1}\mathbf{u}(\mathbf{g})$, where $\mathbf{C}$ is some curvature matrix (often a damped Hessian, Fisher or Gauss-Newton matrix).

It is commonly assumed that first-order methods provide computational efficiency at the inconvenience of manually selecting $\alpha$, while second-order methods suffer a large computational cost to dynamically select an implicit $\alpha$ and improved update direction $\mathbf{d}$ using their more powerful model of the objective. However, a slew of 'adaptive' first-order optimisers (such as Adam (Kingma & Ba, 2015) and relations) blur this distinction by constructing stateful models of the objective function, which can often be interpreted as approximating the curvature of $f(\boldsymbol{\theta})$.

Moreover, practical second-order methods for ML are necessarily approximate, as the curvature $\mathbf{C}$ is otherwise intractably large. Further engineering is then required to mitigate the impact of approximate curvature and the inevitable non-convexity of $f$. For example, in K-FAC, Martens & Grosse (2015) convincingly argue for a particular Kronecker factorisation of a block-diagonal $\mathbf{C}$, but then augment it with a raft of corrections and adaptive heuristics (including multiple periodically-updated damping/factorised Tikhonov regularisation terms, momentum, weight decay, exponential moving averages of curvature statistics and approximate exchange of expectations and Kronecker products). Further, these additions are seemingly *essential* ingredients of a working K-FAC implementation.

A natural question is then whether curvature information or engineering heuristics contribute more to K-FAC's success. In particular, we might ask if accepting first-order methods' inaccurate curvature models and applying second-order stability techniques would blend the computational efficiency and optimisation accuracy of each. Our proposition is thus to adapt Adam using techniques from K-FAC.

### 3.2 ADAM REVISITED

Algorithm 1 restates the Adam optimisation algorithm from Kingma & Ba (2015) applied to $f$, with some minor notational changes. Our proposed algorithm derives from our anecdotal observation that Adam often makes good choices of update direction, which we notate by $\mathbf{d}_t = \frac{\widehat{\mathbf{m}}_t}{\sqrt{\widehat{\mathbf{v}}_t} + \epsilon}$.

As we detail in Appendix C, Adam is known to carry a diagonal approximation to the empirical Fisher matrix in $\widehat{\mathbf{v}}_t$. Then, the $\frac{1}{\sqrt{\widehat{\mathbf{v}}_t} + \epsilon}$ term in Algorithm 1 effectively performs a curvature transformation on the averaged gradient $\widehat{\mathbf{m}}_t$ before computing a more traditional gradient-based update for $\boldsymbol{\theta}$. There are widely-known limitations to using the empirical Fisher in place of the true Fisher information matrix

(Kunstner et al., 2019), and the square root is motivated only by a desire to be "conservative" (Kingma & Ba, 2015). Indeed, Zhang et al. (2018) show Adam is very similar to one construction of natural gradient mean-field variational inference, a technique which is known to prioritise locally fitting modes of the target probability distribution (Turner & Sahani, 2011). The consequent underestimation of global variance corresponds to overestimating local curvature in optimisation, justifying Kingma & Ba (2015)'s preference for a conservative estimate. Nonetheless, this formulation invites us to view Adam through a second-order optimisation lens; we may then ask whether common heuristics applied to second-order optimisers might bring similar benefits to Adam.

### 3.3 Adopting Heuristics from K-FAC

After defining its Kronecker-factored block diagonal approximation to the curvature matrix, K-FAC (Martens & Grosse, 2015) includes three important stabilising heuristics: Levenberg-Marquardt damping, and learning rate and momentum selection according to a local second-order model. Since Adam already implements a momentum correction in $\widehat{\mathbf{m}}_t$, we consider only the first two techniques.

Levenberg-Marquardt damping (Levenberg, 1944; Marquardt, 1963; Roweis, 1996) replaces the curvature matrix $\mathbf{C}$ with the damped version $\mathbf{C}+\lambda\mathbf{I}$, and can variously be interpreted as approximating a trust region, enforcing positive definiteness of $\mathbf{C}$, preventing large updates in low-curvature directions and interpolating between gradient descent and full second-order updates. In effect, it imposes a 'minimum curvature' on the objective to avoid issues from near-zero eigenvalues of $\mathbf{C}$.

Let $M(\boldsymbol{\theta})$ be an approximate second-order model around $\boldsymbol{\theta}_{t-1}$, defined by a truncated Taylor series:

$$M(\boldsymbol{\theta}) = f(\boldsymbol{\theta}_{t-1}) + (\boldsymbol{\theta} - \boldsymbol{\theta}_{t-1})^\mathsf{T}\mathbf{g}_t + \frac{1}{2}(\boldsymbol{\theta} - \boldsymbol{\theta}_{t-1})^\mathsf{T}(\mathbf{C} + \lambda\mathbf{I})(\boldsymbol{\theta} - \boldsymbol{\theta}_{t-1}). \tag{1}$$

The damping parameter $\lambda$ is adapted by comparing the change in objective value predicted by the model $(M(\boldsymbol{\theta}_t) - M(\boldsymbol{\theta}_{t-1}))$ to the actual observed change $(f(\boldsymbol{\theta}_t) - f(\boldsymbol{\theta}_{t-1}))$. This adjustment quantifies the model's reliability by a reduction ratio $\rho$, incorporating stepping factors[1] $\omega_{\text{dec}}, \omega_{\text{inc}}$:

$$\rho = \frac{f(\boldsymbol{\theta}_t) - f(\boldsymbol{\theta}_{t-1})}{M(\boldsymbol{\theta}_t) - M(\boldsymbol{\theta}_{t-1})}; \qquad \lambda \leftarrow \begin{cases} \omega_{\text{dec}}\lambda & \text{if } \rho > \frac{3}{4} \\ \lambda & \text{if } \frac{1}{4} \leq \rho \leq \frac{3}{4} \\ \omega_{\text{inc}}\lambda & \text{if } \rho < \frac{1}{4} \end{cases}. \tag{2}$$

We discuss this formulation further in Appendix A.4.

Once an update direction $\mathbf{d}_t$ has been chosen, a learning rate $\alpha$ is selected according to the second-order model $M$. Specifically, we minimise $M(\boldsymbol{\theta}_{t-1} - \alpha\mathbf{d}_t)$ with respect to $\alpha$, which yields

$$\alpha = \frac{\mathbf{g}_t^\mathsf{T}\mathbf{d}_t}{\mathbf{d}_t^\mathsf{T}(\mathbf{C} + \lambda\mathbf{I})\mathbf{d}_t}. \tag{3}$$

A minor rearrangement shows the large matrix $\mathbf{C}$ only appears in products with vectors. The Jacobian-vector product trick (Pearlmutter, 1994), efficient Fisher decompositions (Martens & Grosse, 2015) and similar techniques compute these quantities using only one additional backward pass per product with $\mathbf{C}$. In practice, the information value of these calculations outweighs this cost.

### 3.4 Extending Adam

Incorporating K-FAC's damping and learning rate selection strategies into Adam yields Algorithm 2, which is easily implementable as a wrapper around vanilla Adam. We name this family of algorithms *AdamQLR*, where *QLR* indicates an optimiser-agnostic quadratic-model learning rate selection logic, which may be applied more broadly (e.g. to SGD).

One remaining consideration is the choice of a curvature matrix $\mathbf{C}$. We use the (true) Fisher matrix throughout, inspired by its connection with Adam's $\widehat{\mathbf{v}}_t$ buffer (see Appendix C.3), its use at the heart of K-FAC and its positive semi-definite guarantee. In short, we tune the damping parameter $\lambda$ to create a trust region in which our quadratic approximation — specified by the Fisher — is

---

[1]In the most general form we allow separate decrease and increase factors, but in practice we will often choose $\omega_{\text{dec}} = \frac{1}{\omega_{\text{inc}}}$ for simplicity. We also require $0 < \omega_{\text{dec}} < 1 < \omega_{\text{inc}}$.

accurate. Then, given the Adam descent direction and the selected $\lambda$, we choose the optimal step size as constrained by this trust region. Our implementation exploits Jacobian-vector products and the efficient Fisher decomposition described in Martens & Grosse (2015, Appendix C), which computes exact products without explicitly storing $\mathbf{C}$.

Finally, our experiments found AdamQLR's training stability to be most threatened by selecting an unreasonably large $\alpha$ for a particular iteration, causing a divergent parameter update. The problem worsens with more model parameters, as this increases the prevalence of low-curvature regions of the space which induce very large update sizes. We found this issue was most effectively mitigated by clipping the learning rate to some maximum $\alpha_{\max}$, and that larger batch sizes tended to improve our curvature estimates, leading to better performance despite the higher cost of each forward pass.

With these choices made, note that the only remaining hyperparameters are $\beta_1$, $\beta_2$ and $\epsilon$ (from Adam) and an initial damping value $\lambda_0$. As it is common for Adam's hyperparameters to be fixed at the default values suggested by Kingma & Ba (2015), and we show $\lambda$ and $\alpha_{\max}$ to be sufficiently insensitive that a default value can be recommended (Section 4.7), we claim that AdamQLR is suitable for use without explicit hyperparameter tuning. In particular, we have encapsulated the learning rate $\alpha$ — arguably the most important hyperparameter to select in many optimisation algorithms. We justify this claim in Section 4.

Compared to Adam, we suffer additional forward and backward passes to compute $M(\boldsymbol{\theta}_t)$ and $(\mathbf{C} + \lambda\mathbf{I})\mathbf{d}_t$. These turn out not to impede performance in our experimental results, though we note a careful implementation would amortise the former cost. Our only significant additional memory cost is storing the vector $(\mathbf{C} + \lambda\mathbf{I})\mathbf{d}_t$, making our approximate memory footprint four times that of SGD (as opposed to Adam's footprint of three times SGD).

## 4 EXPERIMENTS

We examine the training and test performance of AdamQLR in a variety of settings:

**Rosenbrock (1960) Function** $f(x,y) = (a-x)^2 + b(y-x^2)^2$ with $a = 1$ and $b = 100$
**UCI Energy** (Tsanas & Xifara, 2012) on an MLP with one hidden layer of 50 units
**UCI Protein** (Rana, 2013) on an MLP with one hidden layer of 100 units
**Fashion-MNIST** (Xiao et al., 2017) on an MLP with one hidden layer of 50 units
**SVHN** (Netzer et al., 2011) on a ResNet-18 (He et al., 2016)
**CIFAR-10** (Krizhevsky, 2009) on a ResNet-18 (He et al., 2016)

We also demonstrate preliminary scalability to ImageNet in Appendix B.1.3, and a study on Penn Treebank in Appendix B.1.4. On UCI datasets we generate random splits using the same sizes as Gal & Ghahramani (2016) and use MSE loss; otherwise, we separate the standard test set, choose ⅙ (Fashion-MNIST and SVHN) or ¹/₁₀ (CIFAR-10) of the remaining data to form a validation set, and use cross-entropy loss. Code for all our experiments is available at `<redacted>` We compare:

**SGD Minimal** Classical mini-batched stochastic gradient descent, with tuned learning rate
**SGD Full** *SGD Minimal* with additional tuned momentum and weight decay
**Adam** (Kingma & Ba, 2015) with tuned learning rate and fixed defaults for other hyperparameters
**K-FAC** (Martens & Grosse, 2015; Botev & Martens, 2022) with tuned initial damping
**AdamQLR (Tuned)** Algorithm 2, using Fisher curvature for $\mathbf{C}$. We tune initial damping, damping adjustment factors $\omega_{\text{dec}}, \omega_{\text{inc}}$ and learning rate clipping
**AdamQLR (Untuned)** *AdamQLR* with fixed batch size 3 200, initial damping 0.001, $\omega_{\text{dec}} = \frac{1}{\omega_{\text{inc}}} = 0.5$ and learning rate clipping 0.1 (justified by Section 4.7 and Appendix B.2)

Except for the Rosenbrock Function and *AdamQLR (Untuned)*, we also tune a batch size over $\{50, 100, 200, 400, 800, 1\,600, 3\,200\}$. All hyperparameter tuning uses ASHA (Li et al., 2020) over 200 random initialisations, where we target a fixed number of training epochs, subject to a maximum runtime of 15 minutes (only reached for CIFAR-10; see Appendix B.1.6 for experiments using runtime as the primary constraint). For our loss evolution figures, we perform 50 runs using each of the best hyperparameters found (measured by final validation loss), then plot the mean and standard

deviation of the median trends of each of 50 bootstrap samples of the results. Following Botev & Martens (2022), where damping is present we clip it to ensure $\lambda \geq 10^{-8}$. With the exception of the Rosenbrock Function, we give a numerical comparison of the end-of-training statistics in Table 5.

In Appendix B.1.6, we present analogous results where the hyperparameters are tuned to minimse training or validation losses after a fixed runtime, without constraining the number of epochs.

## 4.1 Rosenbrock Function

The Rosenbrock Function (Rosenbrock, 1960) provides a visualisable low-dimensional test bed for optimisation algorithms, containing substantial non-linear correlations between its inputs and anisotropic curvature. We consider 200 optimisation steps, using $\mathcal{N}(\mathbf{0}, \mathbf{I})$-sampled initial $(x, y)$ values during hyperparameter tuning, and plot trajectories from the fixed starting point $(1, -1)$ as our test case in Figure 1. As there is no probabilistic model, we cannot apply K-FAC in this setting, so omit it. For the same reason, in this section only, we use Hessian curvature in *AdamQLR*, and use gradient descent (*GD*) in place of *SGD*. Since there is no separate validation set, we tune hyperparameters on the same objective function as is used for 'training'.

Here, *GD Minimal* makes good initial progress into the central 'valley', but its learning rate is too small to continue along the valley floor. *GD Full*'s hyperparameters cause it to bounce unstably around the optimisation space. Because SGD cannot adapt to different gradient magnitudes, it must select conservative step sizes to avoid diverging when initialised away from the optimum — an effect particularly pronounced in *GD Minimal*, where there is no momentum buffer. *Adam*'s adaptive buffers allow it to target the valley more directly, eventually making slow progress along the valley floor, but it takes time to learn the new dynamics in the latter regime, and we see it initially 'overshoot' the valley.

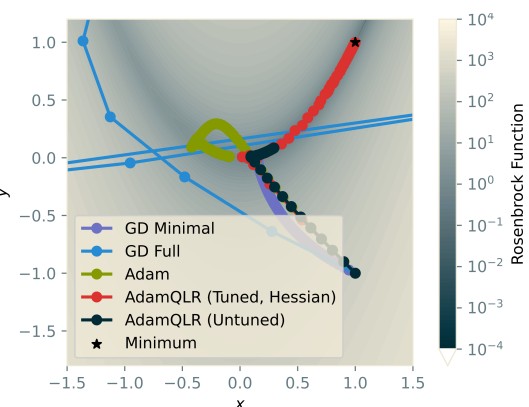

Figure 1: Optimisation trajectories over 200 steps from a fixed initial point on the Rosenbrock Function. Hyperparameter tuning used 200 standard-normal random initial points.

By contrast, *AdamQLR (Tuned)* reaches the valley floor efficiently, then shows an appealing understanding of the objective function geometry, tracking along the valley for substantial distances. SGD-based methods tend to take small, cautious steps along the floor, producing steady but slow convergence, while the Adam-based methods are able to take larger steps, making faster progress. *AdamQLR (Untuned)*'s learning rate clipping threshold, being chosen for neural network applications, is too small here, but it also makes efficient progress into the valley and quickly adapts to the changing dynamics without overshooting. While this relatively simple function is not representative of the more complicated spaces of machine learning model parameters, our strategy displays a promising understanding of its correlated curvature.

## 4.2 UCI Energy

UCI Energy provides a low-dimensional regression task on a small dataset, which is amenable to hosting long experiments to explore convergence behaviour. We consider 4 000 epochs of training and plot bootstrap-sampled median training and test loss trends in Figure 2a.

Our principal benchmarks fall much as we would expect: *SGD Minimal* makes respectable, if sluggish, progress during optimisation, but is outclassed by the more rapid initial convergence of *SGD Full* and *Adam*. Both these latter methods achieve strong test performance on this small-scale problem, with *SGD Full* outperforming all other methods. Despite making rapid initial progress, *K-FAC* quickly begins overfitting, reaching a final test loss similar to the *AdamQLR* methods.

Generally, *AdamQLR (Tuned)* and *(Untuned)* compete comparably with their vanilla baseline. The QLR computed learning rates accelerate initial progress, while the addition of damping provides some defence against overfitting, at the cost of a higher final training loss. Note also that *AdamQLR*'s

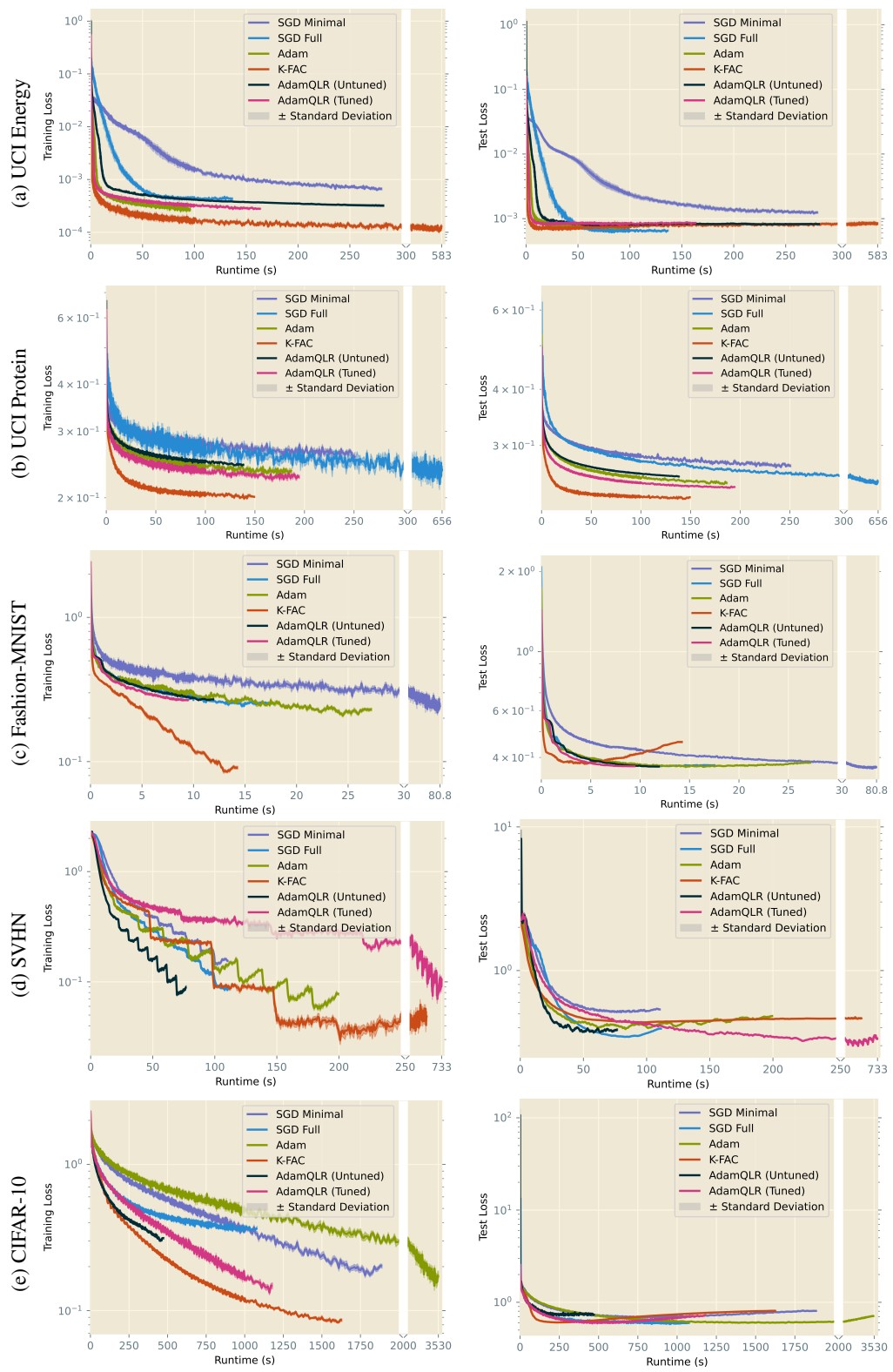

Figure 2: Median training (left) and test (right) loss trajectories, bootstrap-sampled over 50 repetitions per algorithm. Hyperparameters chosen by ASHA over 200 initialisations. Note changes of scale on the time axes. See also results on accuracy metrics and learning rate evolutions in Figures 4 and 5, and numerical comparison in Table 5.

substantially lower variation indicates a robustness beyond that seen in other methods — the *Untuned* variation performs very competitively considering its competition has undergone hyperparameter tuning.

## 4.3  UCI PROTEIN

UCI Protein is another low-dimensional regression task, but with far more data points, allowing for a computationally-efficient study of a larger dataset. We show 200 epochs of training in Figure 2b.

Here we see greater distinction between the generalisation performance of each algorithm. *SGD Full* achieves a slight improvement over *SGD Minimal*, but still lags behind the other methods. *K-FAC* is now clearly the best-performing algorithm, as might perhaps be expected since it computes the most granular curvature approximation when choosing an update direction. However, we still see meaningful benefit from the *AdamQLR* algorithm, with the *(Tuned)* variant now comfortably outperforms *Adam*. We observe *AdamQLR*'s automatic learning rate selection is capable of outperforming methods which require a sensitive explicit choice of learning rate — the *Untuned* variant is clearly superior to tuned *SGD* on this task and is only slightly worse than a tuned *Adam*.

## 4.4  FASHION-MNIST

Fashion-MNIST provides a first foray into higher-dimensional data, but at a scale still approachable by MLP models. Using a 10-epoch training window, we plot bootstrapped loss evolutions in Figure 2c and accuracy evolutions in Figure 4a.

At this slightly larger experimental scale, the benefits of our proposed algorithm become more apparent. Despite achieving the best final training loss of any method, *K-FAC* significantly overfits even before reaching other algorithms' final training losses. While this is a recognised issue with K-FAC (Martens et al., 2018), and the fundamental idea of minimising a test loss by optimising a training loss frustrates the application of naïvely-powerful optimisers, the impact is to make *K-FAC* undesirable in this application. *SGD Full*, *Adam* and *AdamQLR* all perform very similarly, generalising better than *K-FAC* and overfitting to a far lesser degree. *AdamQLR* is the most performant algorithm by a very small margin. We emphasise that the number of training epochs was chosen arbitrarily based on existing work, so the flattening-out of *AdamQLR*'s test loss at later times indicates robustness, not preferential treatment. We note again the strong performance of *AdamQLR (Untuned)*.

## 4.5  SVHN

With SVHN, we progress to a full-colour image dataset and a substantially larger-scale ResNet-18 model, which we tune for 10 epochs and present in Figures 2d (losses) and 4b (accuracies). The periodicity in these loss evolutions corresponds to individual epochs, and is simply an artifact of training.

On this more realistically-scaled problem, we achieve substantial gains over *Adam*. *SGD Minimal* fulfils its expected role as a mediocre baseline, but *SGD Full* performs admirably in this setting, matching the other algorithms' initial rate of convergence in both training and test losses, and achieving the lowest test loss of any method. However, it then overfits, while other methods reach similar test losses more stably. *K-FAC* again fails to generalise its impressively low training losses, instead becoming stuck at a test loss almost ten times larger than its final training loss.

We see particularly strong performance from the Adam-based methods. While *Adam* itself overfits before matching its competitors' test performance, *AdamQLR* reaches impressively low test losses and remains more stable there. Even though *SGD Full* transiently achieves better performance, *AdamQLR* is a more promising candidate for general application, as it achieves similar losses with greater robustness and meaningfully reduced hyperparameter tuning effort. Additionally, the *Untuned* variant performs impressively at both training- and test-time, reinforcing its efficiency and utility.

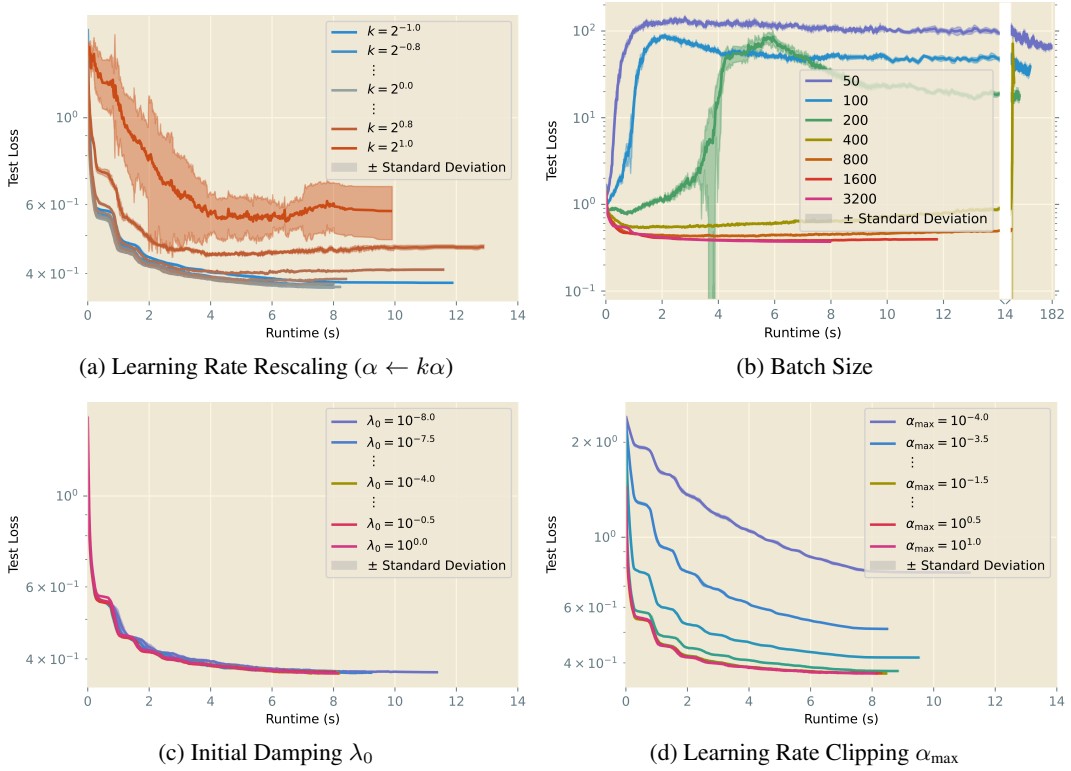

(a) Learning Rate Rescaling ($\alpha \leftarrow k\alpha$)

(b) Batch Size

(c) Initial Damping $\lambda_0$

(d) Learning Rate Clipping $\alpha_{\max}$

Figure 3: Sensitivity studies for *AdamQLR* on Fashion-MNIST over (a) learning rate rescaling, (b) batch size, (c) initial damping and (d) learning rate clipping, showing test losses.

## 4.6  CIFAR-10

Finally, in a simulation of larger-scale learning, we train a ResNet-18 on CIFAR-10 over 72 epochs. Here we include conventional data augmentation of 4-pixel padding, random cropping and random left-right flipping, displaying our loss results in Figure 2e and accuracy results in Figure 4c.

*Adam* is now slower to converge in both training and test loss, suggesting this could be an ill-suited setting in which Adam can be expected to underperform (Balles & Hennig, 2018). Otherwise, increasingly intricate algorithms make progressively faster progress at training-time, even if the generalisation performances are all very similar. The latter effect may reflect inherent issues in the training-test learning paradigm as well as the performance of any particular optimiser.

## 4.7  SENSITIVITY STUDIES

In Appendix B.2 we analyse the sensitivity of *AdamQLR* on Fashion-MNIST by repeating the experiments of Section 4.4 with a range of batch sizes, learning rate clipping thresholds, initial damping values and damping adjustment factors, and by replacing the approximately-optimal learning rate $\alpha$ from (3) with the rescaled $k\alpha$, for various $k \in [0.5, 2.0]$. Figure 3 summarises our results under our standard bootstrapping methodology for each intervention.

Our results inspire further confidence in AdamQLR. Generalisation performance is optimised by choosing $k \approx 1$: constant rescaling of our proposed learning rates does not reduce test error, suggesting we adapt well to the local space and select performant update magnitudes for each direction $\mathbf{d}_t$ proposed by Adam. By contrast, AdamQLR is insensitive to the choice of initial damping $\lambda_0$ on this dataset, so while our ablation studies in Section B.3.1 indicate damping is an important stabilising feature of our method, it appears the adaptive strategy of (2) selects an appropriate damping magnitude regardless of its starting point. While learning rate clipping is not of prime importance in the Fashion-MNIST setting, we verify the expected effect of changing the threshold $\alpha_{\max}$. Finally, larger batch sizes increase generalisation performance. Since we depend

implicitly on highly-parameterised curvature matrices, larger batch sizes would be expected to give a more performant average, but this also substantially decreases training time, owing to efficient GPU computation. All these results justify our *AdamQLR (Untuned)* hyperparameter choices.

## 5   CONCLUSION

In this paper we propose AdamQLR, an extension to Adam which borrows learning rate selection and adaptive damping strategies from second-order methods. Empirically, our algorithm reduces the overfitting seen in other techniques such as K-FAC, is robust to its hyperparameters and is competitive with methods which require tuning of learning rates. Further, an untuned version of AdamQLR, motivated by our sensitivity results, competes with tuned implementations of popular algorithms. Indeed, our observation that AdamQLR competes so strongly with K-FAC, despite representing an algorithmic 'midpoint' between Adam and K-FAC, provides an interesting direction for future work.

We note challenging training-test dynamics from the CIFAR-10 results which merit further investigation, though we leave this to future work. Ultimately, we would like to better understand the workings of second-order methods like K-FAC, such that we can unify the benefits of first- and second-order optimisation to better serve the needs of the ML community, since these significantly differ from those of other optimisation practitioners. In future work, we hope to advance this line of research and better address this fundamental component of ML systems.

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

# A  NOTES

## A.1  ETHICS STATEMENT

Our work proposes a general optimisation algorithm for neural networks, so is unlikely to influence a particular societal problem. However, increasing the effectiveness of optimisation methods makes it easier for both benevolent and malevolent actors to develop systems aligned with their goals, so this class of risk is unavoidable. Of additional concern is that the typical setting of seeking to optimise a test metric by minimising a training metric fundamentally misaligns our algorithms with our objectives, and that misalignment may cause unexpected downstream consequences if poorly understood by the model developer. Finally, it would be naïve to presume any one optimisation algorithm is a panacea for all settings, and any errant belief in this vein may cause promising research directions to be incorrectly dismissed if a supposedly 'universal' optimiser happens to perform poorly on it.

## A.2  REPRODUCIBILITY STATEMENT

We describe our algorithm fully in Section 3, provide full source code to the reviewers and will publish this code to the community after deanonymisation. The descriptions in this paper describe all the modifications we make to Adam and provide a complete intuitive summary of our contribution, while the source code allows any fine detail of our implementation or experiments to be inspected.

## A.3  LIMITATIONS

While we have evaluated our algorithm on a range of datasets and models, we have necessarily left many important settings untested. Thus, even though we expect our method to generalise well to other settings, we should recognise that it has likely not yet been tested in those settings. In particular, the learning rate selection strategy used by K-FAC and our work assumes the optimisation space is approximately convex and quadratic, which will not generally be true of machine learning problems — this motivates our use of damping to defend against particularly ill-posed updates. With sufficient damping, we effectively define a 'trust region' beyond which the surface can be non-quadratic without harming our method. Further, since Adam is known not to perform well in certain (poorly-understood) circumstances (Balles & Hennig, 2018), we might expect AdamQLR to have difficulty with the same class of problems.

## A.4  REDUCTION RATIO

Here, we give a more verbose commentary on the damping adjustment mechanism described in (2).

The definition of the reduction ratio $\rho$ is intuitive. When we update the model parameters from $\boldsymbol{\theta}_{t-1}$ to $\boldsymbol{\theta}_t$, we will observe some change in the loss metric $f(\boldsymbol{\theta}_t) - f(\boldsymbol{\theta}_{t-1})$. Similarly, our quadratic model $M$ will have proposed this parameter update predicting the loss will change by $M(\boldsymbol{\theta}_t) - M(\boldsymbol{\theta}_{t-1})$. Ideally, we would like our model to be a good fit for the true optimisation surface, in which case the observed and predicted changes will be similar, and we will find $\rho \approx 1$. Conversely, if the fit is poor, the observation and prediction will be very different, giving $\rho < 1$ if the observed change is much smaller than the model predicted, or $\rho > 1$ if the change is much larger than the model predicted.

If we find the fit of $M$ to be poor, we would like to adjust the damping to help rectify the situation, since a larger damping will generally bias the model towards expecting larger loss changes, thus proposing smaller parameter updates. Broadly speaking, $\rho > 1$ suggests the model is being too conservative, and we would benefit from decreasing damping to better reflect the underlying surface. Conversely, $\rho < 1$ suggests the model expects much more dramatic changes than we actually see, so we should increase damping to 'reign in' the predictive behaviour.

As Martens & Grosse (2015) note in the original presentation of K-FAC, the optimisation dynamics will change during training. In particular, as we approach a local minimum, the loss surface becomes more and more quadratic-like. Under these circumstances, damping slows down convergence by reducing parameter update sizes, without achieving any appreciable benefit. Even away from local minima, damping tends to trade convergence speed for stability. In both cases, there is a natural incentive to be biased towards decreasing damping if at all possible.

Table 1: Hyperparameter search spaces for Section 4

| Hyperparameter | Search Range |
|---:|:---|
| Batch Size | Uniform in $\{50, 100, 200, 400, 800, 1\,600, 3\,200\}$ |
| Learning Rate $\alpha$ | *SGD*: Logarithmic in $[10^{-6}, 10^{-1}]$
*Adam*: Logarithmic in $[10^{-6}, 1]$ |
| Learning Rate Clipping $\alpha_{\max}$ | Logarithmic in $[10^{-4}, 10]$ |
| Momentum | Logarithmic in $[10^{-4}, 0.3]$, subtracted from 1 |
| Weight Decay | Logarithmic in $[10^{-10}, 1]$ |
| Initial Damping $\lambda_0$ | Logarithmic in $[10^{-8}, 1]$ |
| Damping Decrease Factor $\omega_{\text{dec}}$ | Logarithmic in $[0.5, 1.0]$ |
| Damping Increase Factor $\omega_{\text{inc}}$ | Logarithmic in $[1.0, 4.0]$ |

In this work, we retain Martens & Grosse (2015)'s damping adjustment thresholds of $\rho > 3/4$ and $\rho < 1/4$, since these choices led to desirable performance from K-FAC. Martens & Grosse articulate their preference for reducing $\lambda$ if possible, and we can understand their chosen thresholds in that light. It is for this reason that the thresholds are not centred about $\rho = 1$, as might have been our intuitive expectation.

## A.5 Hyperparameter Search Space

We use similar hyperparameter search spaces (with unused hyperparameters removed) for each dataset and algorithm combination. These are detailed in Table 1.

## A.6 Chosen Hyperparameters

The best hyperparameters selected by ASHA for each setting considered in this work are indicated in Table 2.

## A.7 Compute Used

Our experiments were performed on one of the two sets of hardware shown in Table 3. All runtime comparisons were performed on like-for-like hardware. We make use of GPU acceleration throughout, with the JAX (Bradbury et al., 2018), Haiku (Hennigan et al., 2020) and KFAC-JAX (Botev & Martens, 2022) libraries, along with various related components of the DeepMind JAX Ecosystem (Babuschkin et al., 2020).

Producing experimental data for every plot in this paper required approximately 228.3 GPU-hours on the Local Cluster and 9.5 GPU-hours on the Consumer Desktop. This accounts for performing multiple trials in parallel on the same GPU where capacity exists and for hyperparameter search, but excludes development, debugging and unit testing time, which would substantially increase these figures.

## A.8 Datasets

The datasets we use are all standard in the ML literature; we outline their usage conditions in Table 4.

## B Additional Experiments

### B.1 Algorithm Comparisons

In this Section, we provide some additional viewpoints into our main results of Section 4.

Table 2: Optimal hyperparameters used to produce the results of Section 4.4

| Dataset | Algorithm | Batch Size | Learning Rate | Learning Rate Clipping | Momentum | Weight Decay | Initial Damping | Damping Decrease Factor | Damping Increase Factor |
|---|---|---|---|---|---|---|---|---|---|
| Rosenbrock | GD Minimal | — | — | — | — | — | — | — | — |
| | GD Full | — | — | — | — | — | — | — | — |
| | Adam | — | $9.8848\times10^{-2}$ | — | — | — | — | — | — |
| | AdamQLR (Tuned, Hessian) | — | — | 6.098 | — | — | $3.0270\times10^{-6}$ | 0.9 | 2.1 |
| | AdamQLR (Untuned) | — | — | 0.100 | — | — | $1.0000\times10^{-3}$ | 0.5 | 2.0 |
| UCI Energy | SGD Minimal | 100 | $9.8838\times10^{-2}$ | — | — | — | — | — | — |
| | SGD Full | 400 | $6.9156\times10^{-2}$ | — | 0.9962 | $1.2866\times10^{-4}$ | — | — | — |
| | Adam | 800 | $2.9913\times10^{-2}$ | — | — | — | — | — | — |
| | K-FAC | 50 | — | — | — | — | $1.0047\times10^{-2}$ | — | — |
| | AdamQLR (Untuned) | 3200 | — | 0.100 | — | — | $1.0000\times10^{-3}$ | 0.5 | 2.0 |
| | AdamQLR (Tuned) | 400 | — | 2.843 | — | — | $8.7094\times10^{-2}$ | 0.5 | 2.3 |
| UCI Protein | SGD Minimal | 400 | $7.0021\times10^{-2}$ | — | — | — | — | — | — |
| | SGD Full | 100 | $2.1694\times10^{-4}$ | — | 0.9970 | $1.5361\times10^{-8}$ | — | — | — |
| | Adam | 800 | $5.4189\times10^{-3}$ | — | — | — | — | — | — |
| | K-FAC | 3200 | — | — | — | — | $2.1064\times10^{-1}$ | — | — |
| | AdamQLR (Untuned) | 3200 | — | 0.100 | — | — | $1.0000\times10^{-3}$ | 0.5 | 2.0 |
| | AdamQLR (Tuned) | 800 | — | 0.141 | — | — | $1.5054\times10^{-4}$ | 0.5 | 1.9 |
| Fashion-MNIST | SGD Minimal | 100 | $8.0075\times10^{-2}$ | — | — | — | — | — | — |
| | SGD Full | 800 | $5.8068\times10^{-2}$ | — | 0.9289 | $1.6522\times10^{-8}$ | — | — | — |
| | Adam | 400 | $2.5634\times10^{-3}$ | — | — | — | — | — | — |
| | K-FAC | 3200 | — | — | — | — | $1.9224\times10^{-1}$ | — | — |
| | AdamQLR (Tuned, Hessian) | 3200 | — | 0.269 | — | — | $2.5420\times10^{-5}$ | 1.0 | 2.8 |
| | AdamQLR (Untuned) | 3200 | — | 0.100 | — | — | $1.0000\times10^{-3}$ | 0.5 | 2.0 |
| | AdamQLR (Undamped) | 3200 | — | 0.149 | — | — | — | — | — |
| | AdamQLR (Tuned) | 3200 | — | 0.219 | — | — | $4.9595\times10^{-3}$ | 0.6 | 1.3 |
| CIFAR-10 | SGD Minimal | 200 | $3.4672\times10^{-2}$ | — | — | — | — | — | — |
| | SGD Full | 400 | $3.8337\times10^{-2}$ | — | 0.9203 | $8.7353\times10^{-4}$ | — | — | — |
| | Adam | 100 | $2.0380\times10^{-4}$ | — | — | — | — | — | — |
| | K-FAC | 1600 | — | — | — | — | $9.0326\times10^{-1}$ | — | — |
| | AdamQLR (Tuned, Hessian) | 200 | — | 0.001 | — | — | $2.1848\times10^{-4}$ | 0.5 | 2.1 |
| | AdamQLR (Untuned) | 3200 | — | 0.100 | — | — | $1.0000\times10^{-3}$ | 0.5 | 2.0 |
| | AdamQLR (Undamped) | 200 | — | 0.001 | — | — | — | — | — |
| | AdamQLR (Tuned) | 400 | — | 0.001 | — | — | $7.1607\times10^{-6}$ | 0.5 | 1.2 |
| SVHN | SGD Minimal | 1600 | $3.8629\times10^{-2}$ | — | — | — | — | — | — |
| | SGD Full | 1600 | $6.0953\times10^{-3}$ | — | 0.9862 | $8.6112\times10^{-7}$ | — | — | — |
| | Adam | 800 | $4.1027\times10^{-4}$ | — | — | — | — | — | — |
| | K-FAC | 800 | — | — | — | — | $6.4013\times10^{-1}$ | — | — |
| | AdamQLR (Untuned) | 3200 | — | 0.100 | — | — | $1.0000\times10^{-3}$ | 0.5 | 2.0 |
| | AdamQLR (Tuned) | 200 | — | 0.001 | — | — | $2.6287\times10^{-8}$ | 0.7 | 1.3 |

Table 3: System configurations used to run our experiments.

| Type | CPU | GPU (NVIDIA) | Python | JAX | CUDA | cuDNN |
|---|---|---|---|---|---|---|
| Consumer Desktop | Intel Core i7-3930K | RTX 2080GTX | 3.10.11 | 0.3.25 | 11.4 | 8.05 |
| Local Cluster | Intel Core i9-10900X | RTX 2080GTX | 3.10.11 | 0.3.25 | 11.8 | 8.05 |

Table 4: Licences under which we use datasets in this work

| Dataset | Licence | Source | Input | Output | Total Size |
|---|---|---|---|---|---|
| UCI Energy | Creative Commons Attribution 4.0 International (CC BY 4.0) | Tsanas & Xifara (2012); Gal & Ghahramani (2016) | 8-Vector | Scalar | 692 |
| UCI Protein | None specified | Rana (2013); Gal & Ghahramani (2016) | 9-Vector | Scalar | 45 730 |
| Fashion-MNIST | MIT | Xiao et al. (2017) | $28 \times 28$ Image | Class (from 10) | 60 000 |
| CIFAR-10 | None specified | Krizhevsky (2009) | $32 \times 32$ Image | Class (from 10) | 60 000 |
| SVHN | None specified | Netzer et al. (2011) | $32 \times 32$ Image | Class (from 10) | 99 289 |

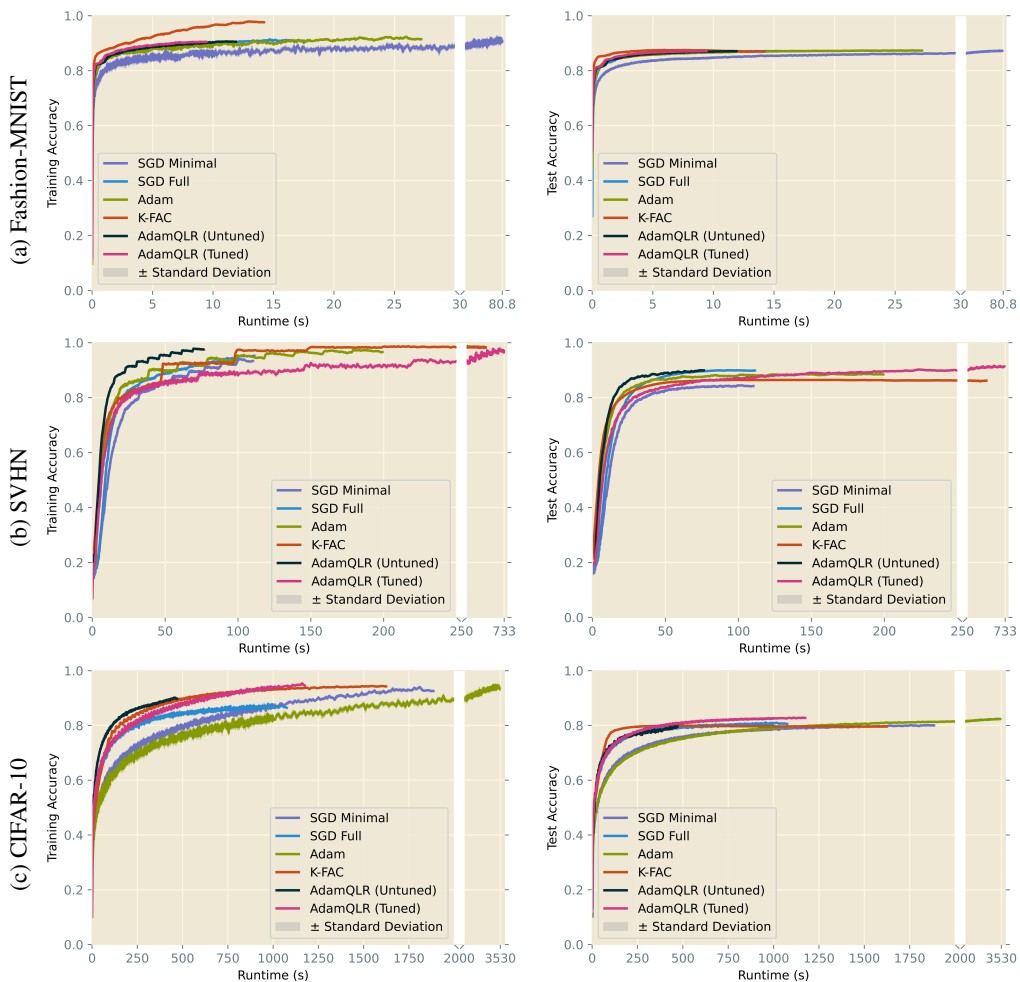

Figure 4: Median training (left) and test (right) accuracy trajectories, bootstrap-sampled over 50 repetitions per algorithm. Hyperparameters chosen by ASHA over 200 initialisations. Note changes of scale on the time axes. See also our numerical comparison in Table 5.

### B.1.1 FASHION-MNIST, SVHN AND CIFAR-10 ACCURACY

In Figure 2, we plotted experimental results in terms of the loss metric used during training. For Fashion-MNIST, SVHN and CIFAR-10, we also present classification accuracy in metrics in Figure 4 and Table 5. These illustrate broadly the same patterns as we discussed in the main body of the paper.

### B.1.2 LEARNING RATE EVOLUTION

In Figure 5, we plot the trajectories of average learning rates selected by AdamQLR and K-FAC against the fixed values used in SGD and Adam.

Learning rate schedules are widely known to be important in certain training problems, particularly at larger scales, so it is unsurprising that various algorithms' sense of the 'optimal' learning rate varies over time. For the most part, the chosen schedules give an approximately exponential decay in learning rate, interestingly excluding the warm-up behaviour commonly specified in manually-designed schedules. In UCI Energy and UCI Protein, we observe a resemblance between the fixed learning rates chosen by SGD and Adam and the typical values selected by AdamQLR and K-FAC, but this connection is much less clear in larger datasets, suggesting this scheduling behaviour becomes more important as problems grow in scale.

Curiously, although *AdamQLR (Tuned)* is able to choose a learning rate clipping value, it only seems to use this to completely disable its adaptive approach — as in SVHN and CIFAR-10 — by setting the threshold lower than the learning rates our QLR strategy would otherwise select. This suggests automatic learning rate selection may not be as useful a tool as we might intuitively think, with a fixed value imposing helpful stability on training. Further, it is interesting to note that *AdamQLR (Untuned)* chooses *growing* learning rates on SVHN and CIFAR-10 which differ dramatically from those of other methods, yet achieves similar results in loss and accuracy space. In summation, these results suggest we might do well to explore other approaches to improving machine learning optimisers, beyond focussing on learning rates.

### B.1.3 IMAGENET

In our explorations, while AdamQLR demonstrated competitive performance with smaller network architectures, its efficacy waned when scaling to larger models, specifically with a ResNet-50 (He et al., 2016) applied to the ImageNet classification task. We adopt the model and accuracy-time evaluation strategy from Dahl et al. (2023) to shed light on these discrepancies, using untuned *Adam* and *AdamQLR* baselines alongside their 'SGD + Heavy ball momentum' setting (which we call *SGD-ImageNet*

From our preliminary plots of training and test accuracy over time in Figure 6, at the initial phase, the performance hierarchy stands as *Adam > AdamQLR > SGD-ImageNet*. However, as training progresses, *AdamQLR* plateaus at a training accuracy of around $70\%$ and a test accuracy of around $50\%$. Unlike *Adam* and *SGD-ImageNet*, which continue their ascent, our method stagnates, unable to further optimise.

A primary reason for this stagnation is the non-convergence of the learning rate in the later stages of training. The algorithm, designed to compute an optimal learning rate for every update step, fails to decrease this rate as training advances, resulting in the persistent large learning rates. This phenomenon suggests that the Levenberg-Marquardt rule, which we employed for damping updates, failed to adjust its damping values for the tail-end of the training process. Future iterations of our algorithm might benefit from a more adaptive damping update mechanism to ensure smoother learning rate annealing. Another intrinsic challenge with our approach is the computation of the optimal learning rate at each step, which requires one evaluation of Fisher vector product per step. For expansive models like ResNet-50, this operation introduces a non-trivial computational overhead. The marginal gains in performance, as observed in our experiments, do not sufficiently offset the increased computational costs for these larger models.

While our method, AdamQLR, introduces promising improvements for certain scenarios, its application to larger networks, like ResNet-50 on ImageNet, surfaces limitations that warrant further research and refinement. We believe that addressing these highlighted challenges can pave the way for a more universally robust optimisation strategy.

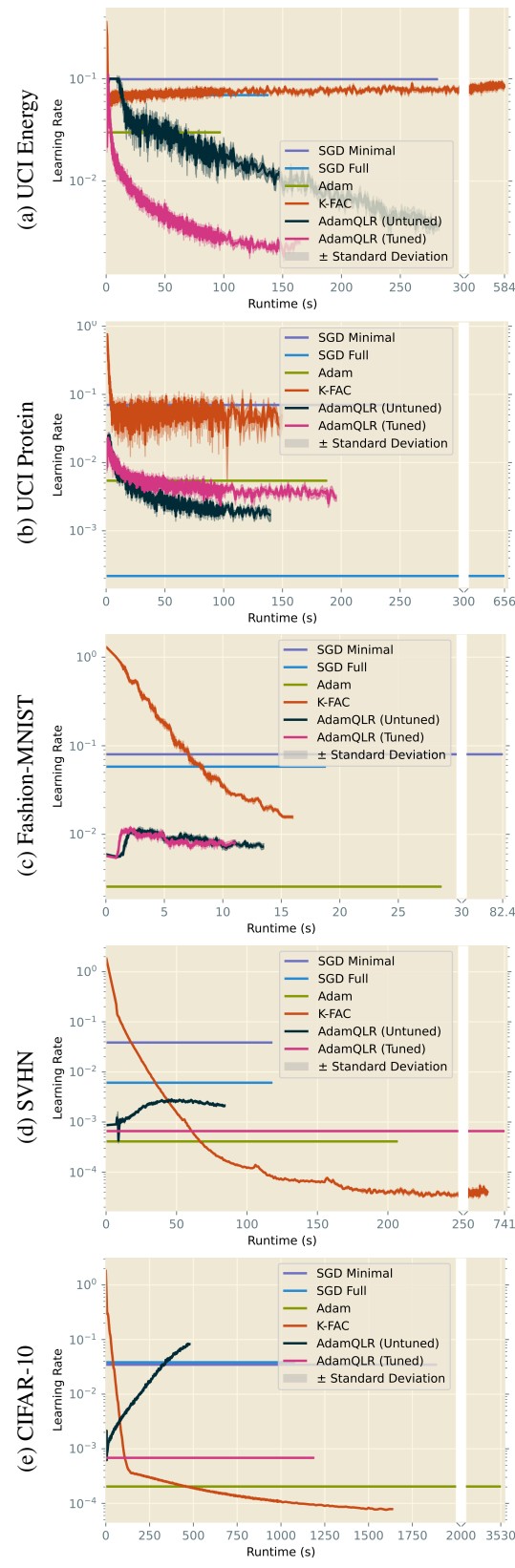

Figure 5: Median learning rate trajectories, bootstrap-sampled over 50 repetitions per algorithm. Hyperparameters chosen by ASHA over 200 initialisations. Note changes of scale on the time axes. See also our numerical presentation in Table 5.

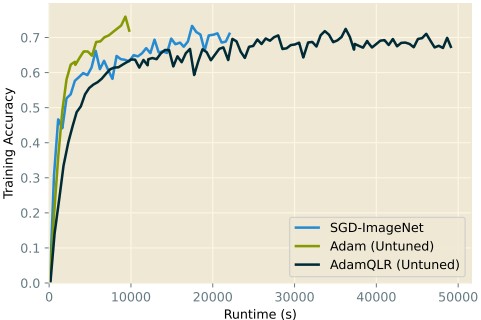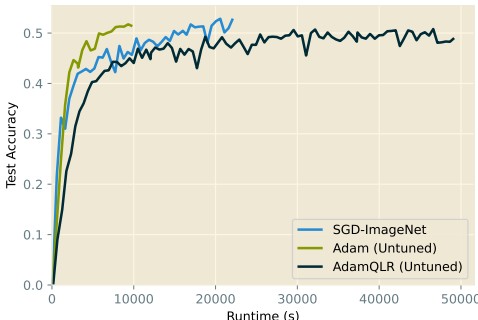

Figure 6: Training (left) and test (right) accuracy vs total training time with ResNet-50 on ImageNet

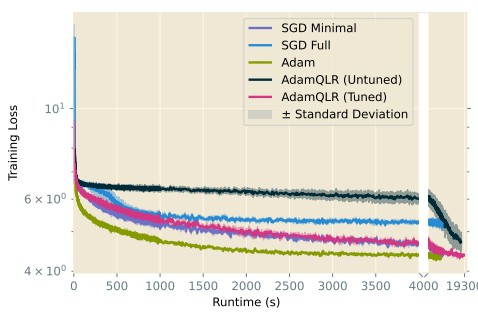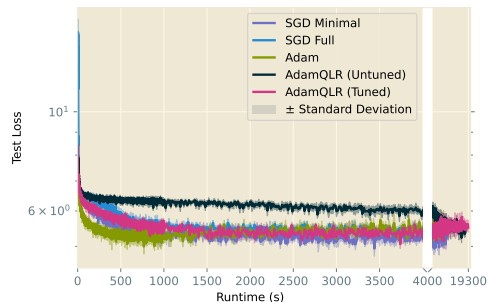

Figure 7: Median training (left) and test (right) loss trajectories for Penn Treebank on GPT-2, bootstrap-sampled over 10 repetitions per algorithm. Hyperparameters chosen by ASHA over 200 initialisations. Note changes of scale on the time axes. See also our numerical presentation in Table 5.

### B.1.4 PENN TREEBANK

As an additional baseline, we consider training the standard Penn Treebank subset (Marcus, Mitchell P. et al., 1999; Marcus et al., 1999) on the GPT-2 model (Radford et al., 2019), as implemented by Hugging Face. We interpret the batch size as the number of token subsequences considered in parallel, chosen over $\{5, 10, 20, 35, 50, 100, 200\}$, and also choose the length of subsequences considered from $\{10, 20, 30, 40, 50, 60, 70, 80, 90, 100\}$, and we set the HPO runtime limit to 1 hour. Otherwise, our hyperparameter optimisation is identical to that of Section 4. For time efficiency, we perform 10 repetitions of training with the best hyperparameters found, rather than 50 as in Section 4, with each repetition comprising 100 epochs of training, and show our results in Figure 7 and Table 5. Our *AdamQLR (Untuned)* setting uses a batch size of 30 and subsequence length of 70, chosen based on the largest values which fit on our GPUs.

Interestingly, our results reflect the observation that transformer training dynamics are quite different from other NN model classes. The addition of momentum and weight decay to *SGD Full* seems to hinder it in comparison to *SGD Minimal*, with the latter exhibiting superior training and generalisation performance. Both are ultimately beaten by *Adam* on training performance, but the latter shows a greater tendency to overfit, with a gradually increasing test loss after around 1000 s which neither *SGD* algorithm exhibits. During training, the chosen *K-FAC* hyperparameters led to immediate and rapid divergence, so we omit *K-FAC* from these plots for clarity.

*AdamQLR (Tuned)* performs very similarly to *SGD Minimal* on this setting, albeit now with a slight tendency to overfit towards the end of training. Further, it achieves similar final training losses to *Adam*, though the latter reaches these losses much faster. On the other hand, *AdamQLR (Untuned)* shows a much greater distinction from *AdamQLR (Tuned)* here than in our other experiments, suggesting the default hyperparameters we propose are not as immediately applicable to transformer models. However, it is reassuring that this algorithm achieves monotonically decreasing training and test losses — combined with *AdamQLR (Untuned)*'s robustness on other experiments, this leads us to suspect that a transformer-specific choice of default hyperparameters would provide similar robustness in this setting. We leave an investigation of these alternative defaults to future work.

Table 5: Numerical study of the results shown in Figures 2, 4 and 7: final statistics after epoch-constrained training on our benchmark tasks.

| Dataset | Algorithm | Training Loss | Training Accuracy | Test Loss | Test Accuracy | Generalisation Gap | Total Steps | Total Time (s) |
|---|---|---|---|---|---|---|---|---|
| UCI Energy | SGD Minimal | 0.000677 ±0.000031 | — | 0.001242 ±0.000049 | — | 0.000565 ±0.000080 | 24000±0 | 276.79 ±0.25 |
| | SGD Full | 0.000431 ±0.000019 | — | 0.000658 ±0.000016 | — | 0.000227 ±0.000036 | 8000±0 | 134.53 ±0.36 |
| | Adam | 0.000278 ±0.000026 | — | 0.000766 ±0.000021 | — | 0.000488 ±0.000047 | 4000±0 | 94.05 ±0.12 |
| | K-FAC | 0.0001349 ±0.0000048 | — | 0.000839 ±0.000030 | — | 0.000704 ±0.000035 | 48000±0 | 580.42 ±0.48 |
| | AdamQLR (Untuned) | 0.000319 ±0.000012 | — | 0.000819 ±0.000027 | — | 0.000499 ±0.000039 | 4000±0 | 278.14 ±0.56 |
| | AdamQLR (Tuned) | 0.0002787 ±0.0000082 | — | 0.000860 ±0.000021 | — | 0.000581 ±0.000029 | 8000±0 | 159.55 ±0.13 |
| UCI Protein | SGD Minimal | 0.2578 ±0.0035 | — | 0.2712 ±0.0020 | — | 0.0134 ±0.0055 | 17600±0 | 249.20 ±0.73 |
| | SGD Full | 0.2379 ±0.0088 | — | 0.2500 ±0.0015 | — | 0.012 ±0.010 | 70000±0 | 648.71 ±0.50 |
| | Adam | 0.2376 ±0.0033 | — | 0.2474 ±0.0011 | — | 0.0098 ±0.0044 | 8800±0 | 185.46 ±0.51 |
| | K-FAC | 0.20150 ±0.00096 | — | 0.2299 ±0.0011 | — | 0.0284 ±0.0020 | 2200±0 | 147.04 ±0.69 |
| | AdamQLR (Untuned) | 0.2444 ±0.0014 | — | 0.25595 ±0.00033 | — | 0.0115 ±0.0017 | 2200±0 | 138.076 ±0.057 |
| | AdamQLR (Tuned) | 0.2261 ±0.0023 | — | 0.24241 ±0.00062 | — | 0.0163 ±0.0029 | 8800±0 | 193.17 ±0.12 |
| Fashion-MNIST | SGD Minimal | 0.2449 ±0.0092 | 0.9053 ±0.0060 | 0.3675 ±0.0011 | 0.87232 ±0.00032 | 0.123 ±0.010 | 5000±0 | 79.677 ±0.067 |
| | SGD Full | 0.2521 ±0.0046 | 0.9097 ±0.0022 | 0.3743 ±0.0018 | 0.87093 ±0.00063 | 0.1222 ±0.0064 | 630±0 | 15.25 ±0.29 |
| | Adam | 0.2278 ±0.0065 | 0.9146 ±0.0021 | 0.3819 ±0.0014 | 0.87303 ±0.00074 | 0.1541 ±0.0079 | 1250±0 | 26.285 ±0.057 |
| | K-FAC | 0.0904 ±0.0016 | 0.97600 ±0.00087 | 0.4570 ±0.0026 | 0.86905 ±0.00052 | 0.3666 ±0.0042 | 160±0 | 13.399 ±0.082 |
| | AdamQLR (Untuned) | 0.2698 ±0.0017 | 0.90517 ±0.00070 | 0.36949 ±0.00057 | 0.87066 ±0.00030 | 0.0996 ±0.0023 | 160±0 | 11.125 ±0.084 |
| | AdamQLR (Tuned) | 0.2670 ±0.0020 | 0.90476 ±0.00088 | 0.37043 ±0.00064 | 0.87078 ±0.00035 | 0.1035 ±0.0027 | 160±0 | 8.47 ±0.12 |
| SVHN | SGD Minimal | 0.1545 ±0.0026 | 0.9336 ±0.0012 | 0.5345 ±0.0042 | 0.84250 ±0.00089 | 0.3799 ±0.0068 | 390±0 | 110.32 ±0.19 |
| | SGD Full | 0.0869 ±0.0020 | 0.95226 ±0.00045 | 0.3945 ±0.0027 | 0.89879 ±0.00053 | 0.3076 ±0.0047 | 390±0 | 110.51 ±0.13 |
| | Adam | 0.0777 ±0.0026 | 0.96620 ±0.00087 | 0.4833 ±0.0030 | 0.88423 ±0.00085 | 0.4056 ±0.0056 | 770±0 | 198.49 ±0.12 |
| | K-FAC | 0.0478 ±0.0045 | 0.9830 ±0.0013 | 0.4641 ±0.0034 | 0.86266 ±0.00077 | 0.4163 ±0.0079 | 770±0 | 500.8 ±1.2 |
| | AdamQLR (Untuned) | 0.0893 ±0.0021 | 0.97478 ±0.00084 | 0.3872 ±0.0061 | 0.8992 ±0.0011 | 0.2980 ±0.0082 | 200±0 | 76.590 ±0.039 |
| | AdamQLR (Tuned) | 0.1077 ±0.0029 | 0.9675 ±0.0030 | 0.3397 ±0.0027 | 0.91182 ±0.00029 | 0.2320 ±0.0056 | 3060±0 | 731.718 ±0.097 |
| CIFAR-10 | SGD Minimal | 0.2019 ±0.0053 | 0.9271 ±0.0033 | 0.8080 ±0.0037 | 0.80012 ±0.00090 | 0.6061 ±0.0090 | 16200±0 | 1846 ±11 |
| | SGD Full | 0.3684 ±0.0083 | 0.8634 ±0.0031 | 0.5977 ±0.0069 | 0.8061 ±0.0021 | 0.229 ±0.015 | 8136±0 | 1058.4 ±2.8 |
| | Adam | 0.173 ±0.010 | 0.9322 ±0.0056 | 0.7093 ±0.0030 | 0.82367 ±0.00052 | 0.536 ±0.013 | 32400±0 | 3481.4 ±3.9 |
| | K-FAC | 0.0857 ±0.0017 | 0.94305 ±0.00077 | 0.8081 ±0.0022 | 0.79596 ±0.00065 | 0.7224 ±0.0039 | 2088±0 | 1599 ±15 |
| | AdamQLR (Untuned) | 0.3114 ±0.0031 | 0.8978 ±0.0010 | 0.735 ±0.013 | 0.7950 ±0.0033 | 0.424 ±0.016 | 1080±0 | 462.86 ±0.52 |
| | AdamQLR (Tuned) | 0.1482 ±0.0076 | 0.9461 ±0.0032 | 0.7172 ±0.0049 | 0.82691 ±0.00092 | 0.569 ±0.012 | 8136±0 | 1176.04 ±0.36 |
| Penn Treebank | SGD Minimal | 4.411 ±0.026 | — | 5.39 ±0.12 | — | 0.98 ±0.15 | 51600±0 | 10090 ±190 |
| | SGD Full | 5.180 ±0.047 | — | 5.338 ±0.064 | — | 0.16 ±0.11 | 53100±0 | 10070 ±150 |
| | Adam | 4.310 ±0.024 | — | 5.54 ±0.13 | — | 1.23 ±0.15 | 37100±0 | 9007 ±24 |
| | AdamQLR (Untuned) | 4.72 ±0.24 | — | 5.47 ±0.10 | — | 0.75 ±0.34 | 44200±0 | 17580 ±320 |
| | AdamQLR (Tuned) | 4.377 ±0.023 | — | 5.45 ±0.10 | — | 1.08 ±0.13 | 51600±0 | 18130 ±350 |

### B.1.5 NUMERICAL RESULTS

In Table 5, we give a numerical presentation of the results in Figures 2, their corresponding accuracy plots from Figure 4 (Appendix B.1.1) and our additional Penn Treebank study from Figure 7 (Appendix B.1.4). We use a similar bootstrapping technique to Section 4 to give estimates for typical runtimes and numbers of steps completed.

### B.1.6 FIXED-RUNTIME COMPARISONS

Our main results in Section 4 impose a primary constraint of a fixed number of epochs, with a secondary constraint of a runtime limit. To develop additional context on AdamQLR's performance, we repeat these experiments without the primary number-of-epochs constraint, such that our hyperparameter tuning directly optimises for the best loss attained after the 15 minute runtime limit, and the algorithms are evaluated on the same metric. Figure 8 and Table 6 show results where we optimised for final validation loss, while Figure 9 and Table 7 show results where the hyperparameters were optimised to minimise final *training* loss. This latter setting allows us to compare the naïve power of each optimiser to optimise the given objective in isolation.

These results display an interesting tendency for *K-FAC* to wildly diverge in the later phases of training on Fashion-MNIST, an effect which *AdamQLR* is largely able to avoid. Broadly speaking, *AdamQLR* gives competitive generalisation performance on UCI Energy and UCI Protein in Figure 8, with a more pronounced overfitting behaviour on larger datasets. However, on CIFAR-10 *AdamQLR (Tuned)* achieves the strongest generalisation, and even on SVHN its performance is competitive. We additionally see an effective demonstration of *AdamQLR*'s optimisation power in Figure 9 — although training performance on Fashion-MNIST again lags behind *Adam* in this setting, larger datasets achieve particularly strong training loss evolutions.

### B.2 SENSITIVITY STUDIES

To justify our configurations and further demonstrate the utility of our algorithm, we conduct a range of sensitivity experiments for *AdamQLR (Tuned)* trained on Fashion-MNIST under the same conditions as in Section 4.4. All hyperparameters except for the one under investigation are fixed at the best values found for ASHA in those experiments. Again, our plots show the averages of median trends of bootstrap-sampled sets of 50 repetitions for each configuration considered.

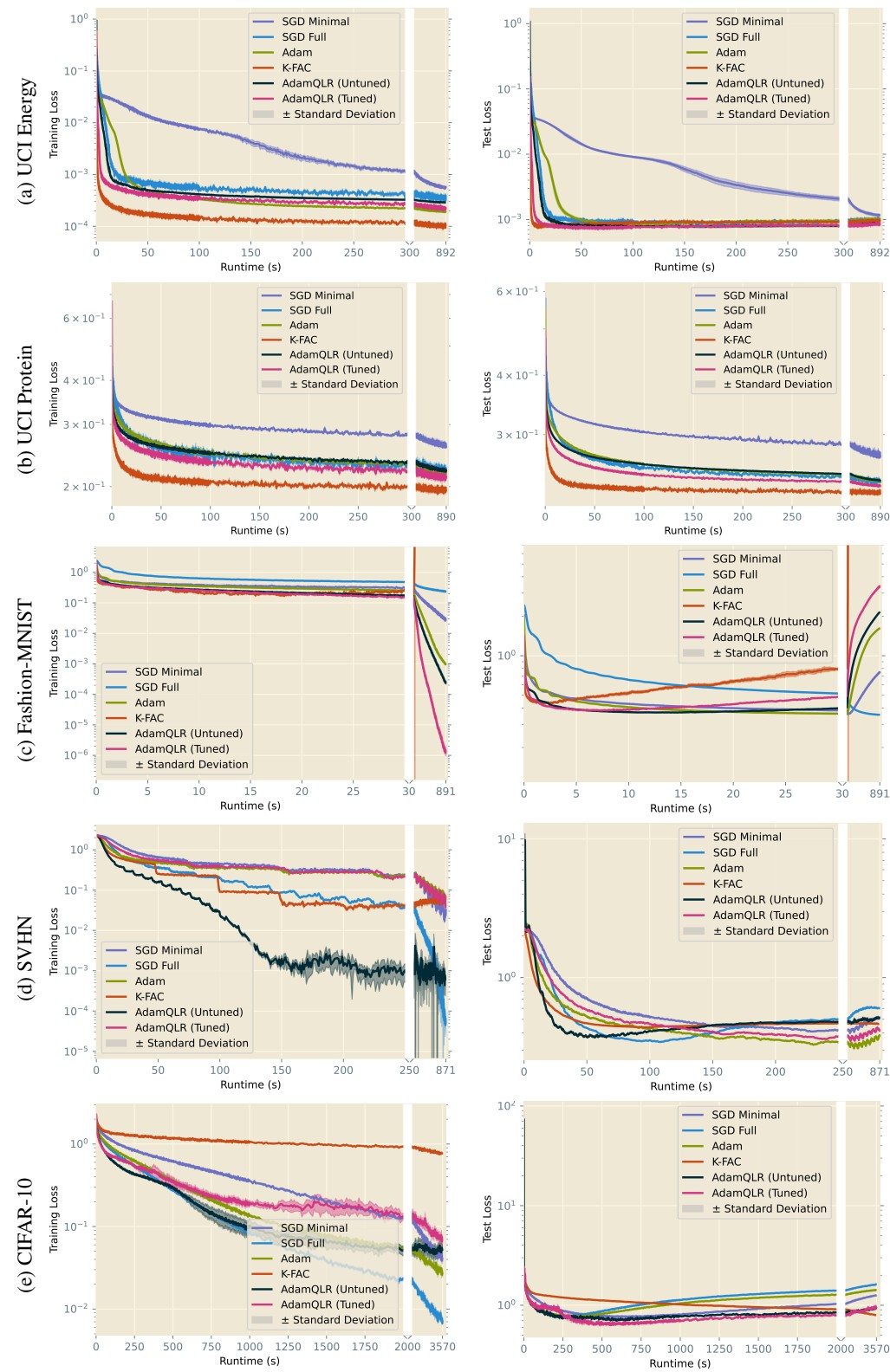

Figure 8: Median training (left) and test (right) loss trajectories, bootstrap-sampled over 50 repetitions per algorithm. Hyperparameters chosen by ASHA over 200 initialisations to minimise validation loss after a fixed runtime of 15 minutes. Note changes of scale on the time axes. See also our numerical comparison in Table 6.

Table 6: Numerical study of the results shown in Figure 8: final statistics after runtime-constrained training on our benchmark tasks, with hyperparameters optimised to minimise *validation* loss.

| Dataset | Algorithm | Training Loss | Training Accuracy | Test Loss | Test Accuracy | Generalisation Gap | Total Steps | Total Time (s) |
|---|---|---|---|---|---|---|---|---|
| UCI Energy | SGD Minimal | $0.000\,554 \pm 0.000\,016$ | — | $0.001\,147 \pm 0.000\,048$ | — | $0.000\,593 \pm 0.000\,064$ | $67\,766 \pm 31$ | $891.221 \pm 0.022$ |
| | SGD Full | $0.000\,385 \pm 0.000\,028$ | — | $0.001\,005 \pm 0.000\,052$ | — | $0.000\,620 \pm 0.000\,080$ | $79\,375 \pm 72$ | $889.608 \pm 0.069$ |
| | Adam | $0.000\,190\,2 \pm 0.000\,004\,3$ | — | $0.000\,996 \pm 0.000\,047$ | — | $0.000\,806 \pm 0.000\,051$ | $38\,460 \pm 120$ | $889.142 \pm 0.072$ |
| | K-FAC | $0.000\,103\,0 \pm 0.000\,006\,3$ | — | $0.000\,906 \pm 0.000\,055$ | — | $0.000\,803 \pm 0.000\,062$ | $67\,973 \pm 72$ | $887.049 \pm 0.059$ |
| | AdamQLR (Untuned) | $0.000\,286 \pm 0.000\,013$ | — | $0.000\,879 \pm 0.000\,046$ | — | $0.000\,593 \pm 0.000\,059$ | $13\,776 \pm 13$ | $888.031 \pm 0.055$ |
| | AdamQLR (Tuned) | $0.000\,215 \pm 0.000\,014$ | — | $0.000\,835 \pm 0.000\,026$ | — | $0.000\,620 \pm 0.000\,040$ | $73\,745 \pm 40$ | $888.765 \pm 0.061$ |
| UCI Protein | SGD Minimal | $0.2602 \pm 0.0025$ | — | $0.2694 \pm 0.0020$ | — | $0.0092 \pm 0.0046$ | $13\,873 \pm 56$ | $887.045 \pm 0.032$ |
| | SGD Full | $0.2208 \pm 0.0057$ | — | $0.239\,29 \pm 0.000\,81$ | — | $0.0185 \pm 0.0065$ | $54\,180 \pm 120$ | $886.933 \pm 0.036$ |
| | Adam | $0.2209 \pm 0.0014$ | — | $0.240\,26 \pm 0.000\,89$ | — | $0.0194 \pm 0.0023$ | $24\,520 \pm 120$ | $886.772 \pm 0.052$ |
| | K-FAC | $0.1948 \pm 0.0018$ | — | $0.225\,70 \pm 0.000\,74$ | — | $0.0309 \pm 0.0026$ | $22\,477 \pm 94$ | $884.766 \pm 0.089$ |
| | AdamQLR (Untuned) | $0.2225 \pm 0.0013$ | — | $0.240\,65 \pm 0.000\,60$ | — | $0.0181 \pm 0.0019$ | $13\,986.2 \pm 5.5$ | $886.118 \pm 0.065$ |
| | AdamQLR (Tuned) | $0.2134 \pm 0.0016$ | — | $0.233\,84 \pm 0.000\,55$ | — | $0.0205 \pm 0.0021$ | $40\,160 \pm 200$ | $886.234 \pm 0.090$ |
| Fashion-MNIST | SGD Minimal | $0.0285 \pm 0.0012$ | $0.9942 \pm 0.0011$ | $0.7482 \pm 0.0027$ | $0.867\,00 \pm 0.000\,30$ | $0.7197 \pm 0.0039$ | $44\,140 \pm 190$ | $886.15 \pm 0.13$ |
| | SGD Full | $0.2343 \pm 0.0035$ | $0.9196 \pm 0.0013$ | $0.355\,15 \pm 0.000\,60$ | $0.875\,04 \pm 0.000\,21$ | $0.1208 \pm 0.0041$ | $20\,942 \pm 29$ | $886.378 \pm 0.066$ |
| | Adam | $0.000\,994 \pm 0.000\,025$ | $1.0 \pm 0$ | $1.622 \pm 0.014$ | $0.860\,59 \pm 0.000\,46$ | $1.621 \pm 0.014$ | $12\,071 \pm 30$ | $886.214 \pm 0.082$ |
| | K-FAC | $3.3 \times 10^9 \pm 1.5 \times 10^9$ | $0.045 \pm 0.017$ | $3.8 \times 10^9 \pm 2.9 \times 10^9$ | $0.1000 \pm 0$ | $6.0 \times 10^8 \pm 4.4 \times 10^9$ | $1884 \pm 48$ | $69.3 \pm 1.9$ |
| | AdamQLR (Untuned) | $0.000\,247 \pm 0.000\,026$ | $1.0 \pm 0$ | $2.146 \pm 0.015$ | $0.857\,94 \pm 0.000\,62$ | $2.146 \pm 0.015$ | $11\,910.6 \pm 8.4$ | $885.49 \pm 0.13$ |
| | AdamQLR (Tuned) | $0.000\,001\,34 \pm 0.000\,000\,39$ | $1.0 \pm 0$ | $3.368 \pm 0.060$ | $0.861\,41 \pm 0.000\,74$ | $3.368 \pm 0.060$ | $20\,790 \pm 100$ | $885.278 \pm 0.051$ |
| SVHN | SGD Minimal | $0.0338 \pm 0.0041$ | $0.9999 \pm 0$ | $0.5213 \pm 0.0011$ | $0.884\,79 \pm 0.000\,46$ | $0.4875 \pm 0.0053$ | $3550 \pm 16$ | $869.934 \pm 0.080$ |
| | SGD Full | $0.000\,060 \pm 0.000\,020$ | $1.0 \pm 0$ | $0.6028 \pm 0.0039$ | $0.913\,64 \pm 0.000\,30$ | $0.6028 \pm 0.0039$ | $3277 \pm 13$ | $867.951 \pm 0.094$ |
| | Adam | $0.0695 \pm 0.0062$ | $0.9791 \pm 0.0024$ | $0.3840 \pm 0.0032$ | $0.910\,35 \pm 0.000\,76$ | $0.3145 \pm 0.0094$ | $3515 \pm 13$ | $868.21 \pm 0.14$ |
| | K-FAC | $0.0522 \pm 0.0032$ | $0.992\,09 \pm 0.000\,94$ | $0.4655 \pm 0.0018$ | $0.861\,02 \pm 0.000\,28$ | $0.4132 \pm 0.0050$ | $1296.0 \pm 4.5$ | $848.46 \pm 0.15$ |
| | AdamQLR (Untuned) | $0.000\,62 \pm 0.000\,32$ | $0.999\,76 \pm 0.000\,12$ | $0.5137 \pm 0.0098$ | $0.918\,31 \pm 0.000\,69$ | $0.513 \pm 0.010$ | $1901 \pm 71$ | $758 \pm 31$ |
| | AdamQLR (Tuned) | $0.0383 \pm 0.0035$ | $0.9876 \pm 0.0023$ | $0.4156 \pm 0.0029$ | $0.901\,87 \pm 0.000\,57$ | $0.3773 \pm 0.0065$ | $3413 \pm 12$ | $865.619 \pm 0.073$ |
| CIFAR-10 | SGD Minimal | $0.0395 \pm 0.0025$ | $0.9868 \pm 0.0010$ | $1.2612 \pm 0.0056$ | $0.777\,25 \pm 0.000\,42$ | $1.2217 \pm 0.0081$ | $26\,182 \pm 66$ | $3567.78 \pm 0.16$ |
| | SGD Full | $0.007\,29 \pm 0.000\,43$ | $0.997\,44 \pm 0.000\,27$ | $1.6215 \pm 0.0024$ | $0.783\,93 \pm 0.000\,72$ | $1.6142 \pm 0.0028$ | $14\,945 \pm 23$ | $3564.70 \pm 0.18$ |
| | Adam | $0.0273 \pm 0.0013$ | $0.989\,91 \pm 0.000\,28$ | $1.4211 \pm 0.0051$ | $0.775\,91 \pm 0.000\,99$ | $1.3938 \pm 0.0063$ | $20\,859 \pm 27$ | $3565.12 \pm 0.14$ |
| | K-FAC | $0.790 \pm 0.017$ | $0.7245 \pm 0.0040$ | $0.7963 \pm 0.0026$ | $0.725\,16 \pm 0.000\,91$ | $0.006 \pm 0.020$ | $8620 \pm 34$ | $3548.36 \pm 0.14$ |
| | AdamQLR (Untuned) | $0.0497 \pm 0.0062$ | $0.9829 \pm 0.0018$ | $0.940 \pm 0.029$ | $0.828 \pm 0.011$ | $0.890 \pm 0.035$ | $8147 \pm 29$ | $3558.55 \pm 0.53$ |
| | AdamQLR (Tuned) | $0.0681 \pm 0.0052$ | $0.978\,94 \pm 0.000\,98$ | $0.966 \pm 0.028$ | $0.8235 \pm 0.0060$ | $0.898 \pm 0.033$ | $13\,029 \pm 11$ | $3560.64 \pm 0.19$ |

Table 7: Numerical study of the results shown in Figure 9: final statistics after runtime-constrained training on our benchmark tasks, with hyperparameters optimised to minimise *training* loss.

| Dataset | Algorithm | Training Loss | Training Accuracy | Test Loss | Test Accuracy | Generalisation Gap | Total Steps | Total Time (s) |
|---|---|---|---|---|---|---|---|---|
| UCI Energy | SGD Minimal | $0.000\,458 \pm 0.000\,016$ | — | $0.001\,030 \pm 0.000\,029$ | — | $0.000\,571 \pm 0.000\,044$ | $68\,024 \pm 24$ | $891.107 \pm 0.025$ |
| | SGD Full | $0.000\,356 \pm 0.000\,017$ | — | $0.000\,950 \pm 0.000\,041$ | — | $0.000\,594 \pm 0.000\,058$ | $53\,931 \pm 27$ | $889.315 \pm 0.045$ |
| | Adam | $0.000\,232 \pm 0.000\,013$ | — | $0.000\,954 \pm 0.000\,027$ | — | $0.000\,721 \pm 0.000\,040$ | $13\,486.1 \pm 6.2$ | $888.042 \pm 0.062$ |
| | K-FAC | $0.000\,271\,8 \pm 0.000\,006\,8$ | — | $0.001\,053 \pm 0.000\,039$ | — | $0.000\,781 \pm 0.000\,045$ | $973 \pm 23$ | $23.57 \pm 0.41$ |
| | AdamQLR (Untuned) | $0.000\,288 \pm 0.000\,012$ | — | $0.000\,875 \pm 0.000\,056$ | — | $0.000\,587 \pm 0.000\,068$ | $13\,777 \pm 11$ | $888.025 \pm 0.056$ |
| | AdamQLR (Tuned) | $0.000\,185\,0 \pm 0.000\,005\,1$ | — | $0.000\,891 \pm 0.000\,029$ | — | $0.000\,706 \pm 0.000\,034$ | $53\,212 \pm 12$ | $888.370 \pm 0.045$ |
| UCI Protein | SGD Minimal | $0.251 \pm 0.010$ | — | $0.2629 \pm 0.0011$ | — | $0.012 \pm 0.011$ | $75\,445 \pm 66$ | $886.867 \pm 0.081$ |
| | SGD Full | $0.2218 \pm 0.0013$ | — | $0.241\,15 \pm 0.000\,49$ | — | $0.0194 \pm 0.0018$ | $13\,812.4 \pm 9.4$ | $886.761 \pm 0.062$ |
| | Adam | $0.2205 \pm 0.0016$ | — | $0.237\,42 \pm 0.000\,67$ | — | $0.0169 \pm 0.0023$ | $39\,108 \pm 16$ | $886.594 \pm 0.053$ |
| | K-FAC | $0.1992 \pm 0.0019$ | — | $0.2265 \pm 0.0012$ | — | $0.0274 \pm 0.0031$ | $22\,382 \pm 16$ | $884.61 \pm 0.11$ |
| | AdamQLR (Untuned) | $0.2230 \pm 0.0012$ | — | $0.240\,62 \pm 0.000\,65$ | — | $0.0177 \pm 0.0018$ | $13\,986.4 \pm 5.3$ | $886.115 \pm 0.070$ |
| | AdamQLR (Tuned) | $0.2133 \pm 0.0012$ | — | $0.234\,30 \pm 0.000\,48$ | — | $0.0210 \pm 0.0048$ | $54\,368 \pm 37$ | $885.834 \pm 0.042$ |
| Fashion-MNIST | SGD Minimal | $0.0191 \pm 0.0011$ | $0.997\,70 \pm 0.000\,64$ | $0.8144 \pm 0.0038$ | $0.865\,82 \pm 0.000\,59$ | $0.7953 \pm 0.0048$ | $43\,829 \pm 64$ | $881.80 \pm 0.77$ |
| | SGD Full | $0.001\,09 \pm 0.000\,26$ | $0.999\,77 \pm 0.000\,15$ | $2.482 \pm 0.047$ | $0.854\,17 \pm 0.000\,54$ | $2.481 \pm 0.047$ | $12\,106.6 \pm 6.2$ | $886.419 \pm 0.058$ |
| | Adam | $0.000\,031 \pm 0.000\,015$ | $1.0 \pm 0$ | $3.27 \pm 0.13$ | $0.865\,31 \pm 0.000\,51$ | $3.27 \pm 0.13$ | $20\,871 \pm 46$ | $885.827 \pm 0.093$ |
| | K-FAC | $6.0 \times 10^{14} \pm 3.5 \times 10^{15}$ | $0.040 \pm 0.012$ | $1.1 \times 10^{14} \pm 3.2 \times 10^{14}$ | $0.1000 \pm 0$ | $-5.0 \times 10^{14} \pm 3.8 \times 10^{15}$ | $5150 \pm 93$ | $439 \pm 11$ |
| | AdamQLR (Untuned) | $0.000\,244 \pm 0.000\,026$ | $1.0 \pm 0$ | $2.139 \pm 0.023$ | $0.857\,73 \pm 0.000\,85$ | $2.139 \pm 0.023$ | $11\,906 \pm 11$ | $885.48 \pm 0.13$ |
| | AdamQLR (Tuned) | $0.000\,249 \pm 0.000\,029$ | $1.0 \pm 0$ | $2.139 \pm 0.028$ | $0.857\,71 \pm 0.000\,63$ | $2.139 \pm 0.028$ | $11\,887.1 \pm 10.0$ | $885.456 \pm 0.093$ |
| SVHN | SGD Minimal | $0.000\,690 \pm 0.000\,019$ | $1.0 \pm 0$ | $0.8493 \pm 0.0053$ | $0.831\,79 \pm 0.000\,53$ | $0.8487 \pm 0.0053$ | $3190.16 \pm 0.37$ | $867.714 \pm 0.068$ |
| | SGD Full | $0.000\,886\,6 \pm 0.000\,004\,4$ | $1.0 \pm 0$ | $0.6512 \pm 0.0028$ | $0.883\,08 \pm 0.000\,47$ | $0.6512 \pm 0.0028$ | $3479.15 \pm 0.59$ | $867.848 \pm 0.090$ |
| | Adam | $0.000\,477\,19 \pm 0.000\,000\,89$ | $1.0 \pm 0$ | $0.9529 \pm 0.0056$ | $0.844\,08 \pm 0.000\,96$ | $0.9529 \pm 0.0056$ | $3154.65 \pm 0.45$ | $865.873 \pm 0.073$ |
| | K-FAC | $0.0249 \pm 0.0024$ | $0.997\,07 \pm 0.000\,39$ | $0.4339 \pm 0.0029$ | $0.8768 \pm 0.0012$ | $0.4090 \pm 0.0053$ | $786.88 \pm 0.41$ | $844.76 \pm 0.15$ |
| | AdamQLR (Untuned) | $0.000\,60 \pm 0.000\,34$ | $0.999\,80 \pm 0.000\,14$ | $0.512 \pm 0.010$ | $0.918\,19 \pm 0.000\,78$ | $0.511 \pm 0.011$ | $1939 \pm 89$ | $773 \pm 40$ |
| | AdamQLR (Tuned) | $0.000\,005\,105 \pm 0.000\,000\,089$ | $1.0 \pm 0$ | $0.6497 \pm 0.0018$ | $0.904\,98 \pm 0.000\,27$ | $0.6497 \pm 0.0018$ | $2234.66 \pm 0.49$ | $861.98 \pm 0.16$ |
| CIFAR-10 | SGD Minimal | $0.1960 \pm 0.0093$ | $0.9257 \pm 0.0054$ | $0.7830 \pm 0.0026$ | $0.796\,29 \pm 0.000\,77$ | $0.587 \pm 0.012$ | $33\,698 \pm 21$ | $3568.649 \pm 0.085$ |
| | SGD Full | $0.026\,56 \pm 0.000\,93$ | $0.992\,28 \pm 0.000\,65$ | $1.2643 \pm 0.0049$ | $0.792\,67 \pm 0.000\,67$ | $1.2377 \pm 0.0058$ | $26\,672.7 \pm 6.4$ | $3566.72 \pm 0.16$ |
| | Adam | $0.0557 \pm 0.0038$ | $0.9819 \pm 0.0019$ | $1.3478 \pm 0.0042$ | $0.756\,02 \pm 0.000\,66$ | $1.2920 \pm 0.0079$ | $26\,696 \pm 10$ | $3565.90 \pm 0.11$ |
| | K-FAC | $1.173 \pm 0.020$ | $0.5800 \pm 0.0048$ | $1.1039 \pm 0.0080$ | $0.6123 \pm 0.0021$ | $-0.070 \pm 0.028$ | $9660.2 \pm 1.7$ | $3548.48 \pm 0.12$ |
| | AdamQLR (Untuned) | $0.0517 \pm 0.0086$ | $0.9826 \pm 0.0024$ | $0.944 \pm 0.030$ | $0.8280 \pm 0.0089$ | $0.893 \pm 0.039$ | $8050 \pm 650$ | $3558.1 \pm 2.7$ |
| | AdamQLR (Tuned) | $0.013\,01 \pm 0.000\,61$ | $0.995\,97 \pm 0.000\,23$ | $1.2872 \pm 0.0066$ | $0.826\,55 \pm 0.000\,71$ | $1.2742 \pm 0.0072$ | $8239 \pm 12$ | $3559.95 \pm 0.15$ |

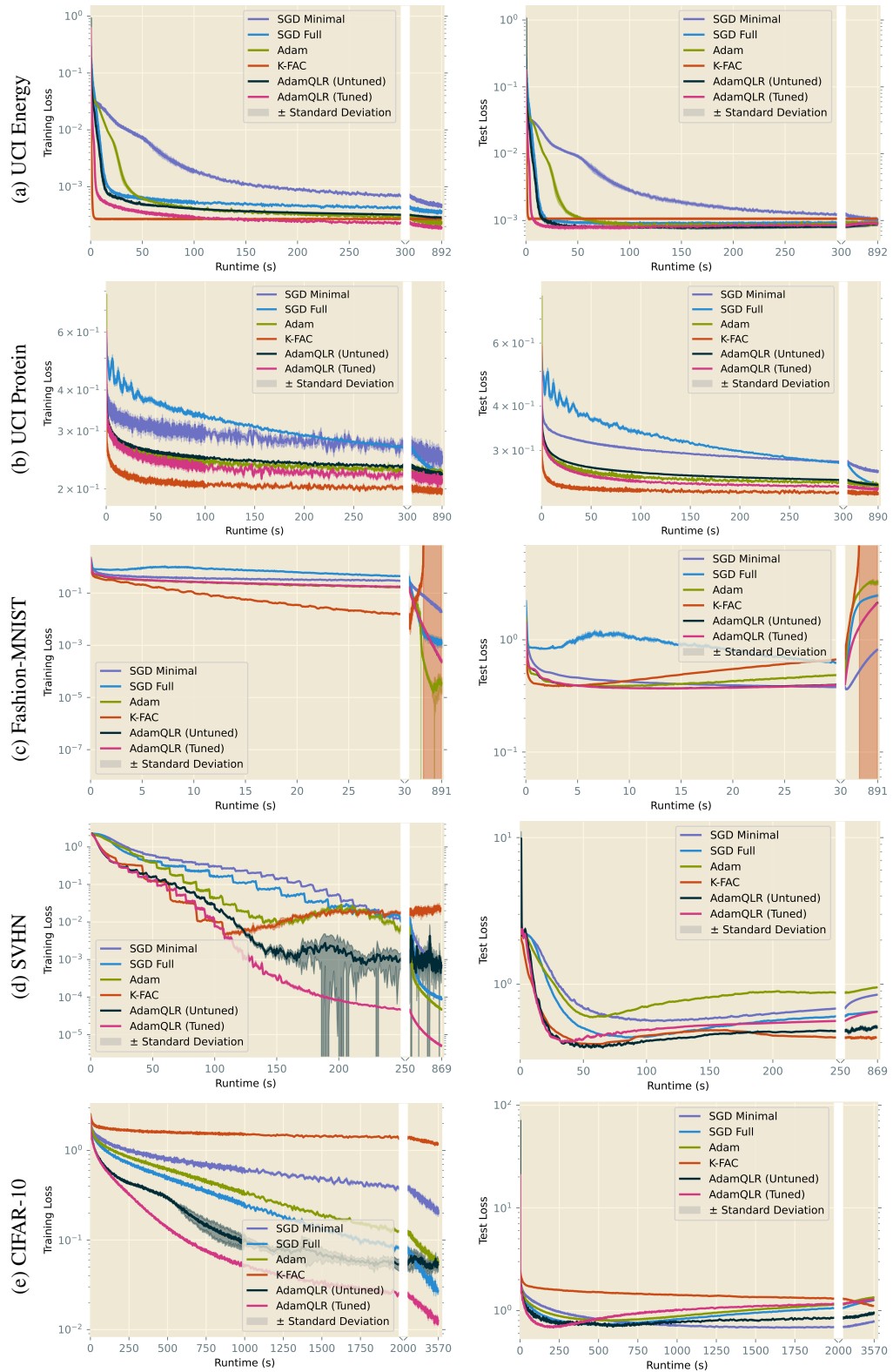

Figure 9: Median training (left) and test (right) loss trajectories, bootstrap-sampled over 50 repetitions per algorithm. Hyperparameters chosen by ASHA over 200 initialisations to minimise *training* loss after a fixed runtime of 15 minutes, characterising the naïve power of each algorithm. Note changes of scale on the time axes. See also our numerical comparison in Table 7

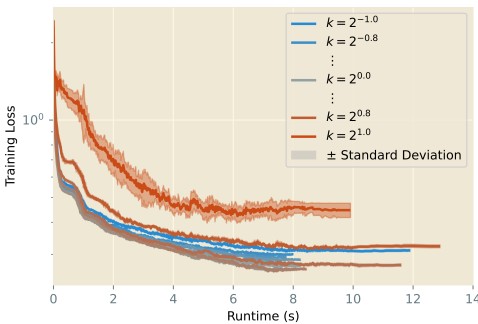 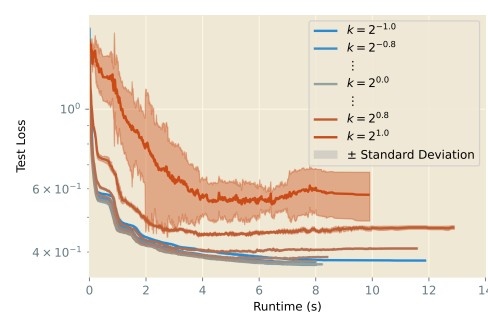

Figure 10: Ablation studies over learning rate, which is scaled by a variety of constant factors $k$ for our Fashion-MNIST trial from Section 4.4.

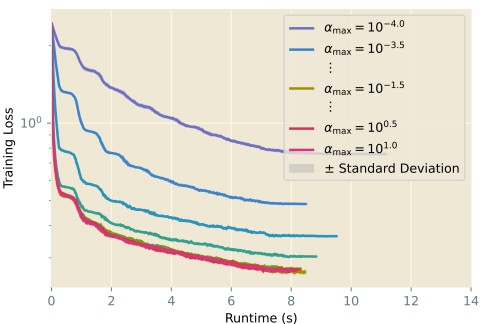 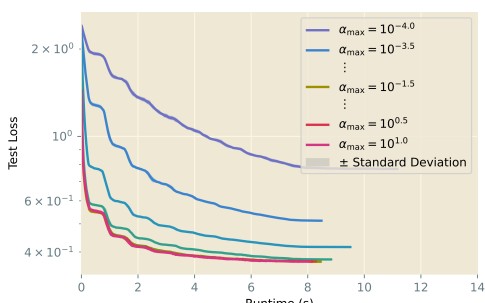

Figure 11: Ablation studies over learning rate clipping $\alpha_{\text{max}}$ for our Fashion-MNIST trial from Section 4.4.

### B.2.1 LEARNING RATE RESCALING

Firstly, we analyse the accuracy of our learning rate selection strategy by executing our algorithm as normal, but setting $\alpha \leftarrow k\alpha$ for each $k$ in $\{2^{-1.0}, 2^{-0.8}, 2^{-0.6}, \cdots, 2^{1.0}\}$. This scaling is performed *after* any clipping has taken place. In effect, we investigate the potential for systemic bias in our learning rate selection by asking if our results would improve with a constant scaling factor on those learning rates.

Our results in Figure 10 show the $k = 2^{1.0}$ case exhibiting large variance due to unstable runs, while the best training losses are obtained for $k$ slightly larger than unity. This makes sense given our use of damping: if stability can be achieved without damping for any given update, then the damping will serve only to downsize our proposed update step, so we should expect the best results to be obtained by slightly increasing it again. However, test loss appears generally less sensitive to $k$, with the lowest value obtained for $k = 1$: this would also be expected under damping, since we would hope the damping would increase generalisation performance. In aggregate, these results confirm our approach accurately selects the correct learning rate to use for any given optimisation step.

### B.2.2 LEARNING RATE CLIPPING

We continue by considering the learning rate clipping threshold $\alpha_{\text{max}}$, selecting values in $\{10^{-4.0}, 10^{-3.5}, 10^{-3.0}, \cdots, 10^{0.0}\}$ and plotting our results in Figure 11.

On Fashion-MNIST, we see a clear preference for a higher learning rate clipping threshold, corresponding to less aggressive clipping, with the effect shrinking after a point as the threshold becomes larger than any learning rate selected by AdamQLR. This makes sense — we introduce learning rate clipping to mitigate the effects of unstably large learning rates, and if these do not arise, we will only harm performance by clipping learning rates. Fashion-MNIST training proceeded successfully without clipping, so this hyperparameter is only of particular importance in larger problems where it is a more vital component of a stable training algorithm. However, it is reassuring to confirm that a

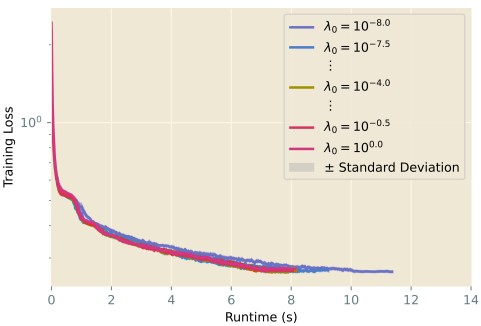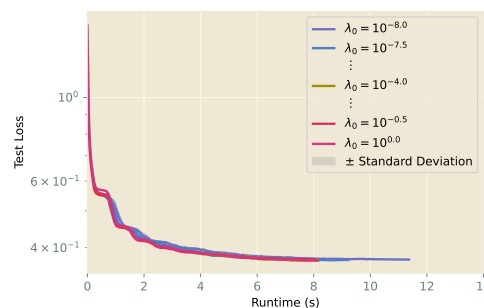

Figure 12: Ablation studies over initial damping value $\lambda_0$ for our Fashion-MNIST trial from Section 4.4.

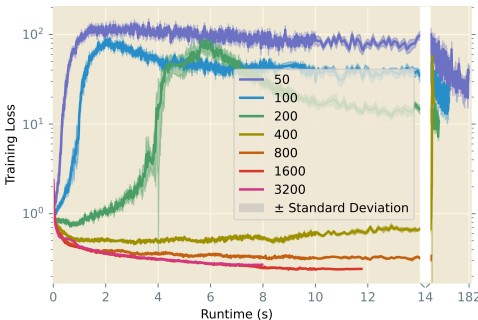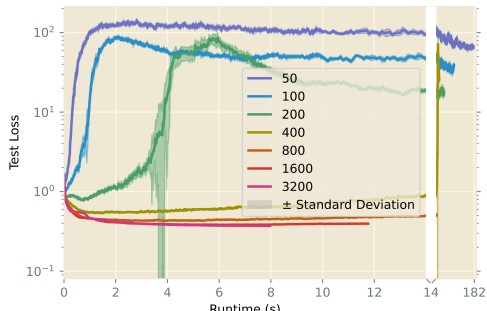

Figure 13: Ablation studies over batch size for our Fashion-MNIST trial from Section 4.4.

sufficiently high learning rate clipping threshold will not drastically harm performance on otherwise stable problems.

### B.2.3 INITIAL DAMPING

Next, we consider the initial value $\lambda_0$ assigned to our Levenberg-Marquardt damping term $\lambda$, testing values in $\{10^{-8.0}, 10^{-7.5}, 10^{-7.0}, \cdots, 10^{0.0}\}$. Here, we seek to quantify the trade-off between damping's stabilising effect and its tendency to worsen training loss. Figure 12 presents our results.

With the exception of the very smallest values, we see our performance is largely insensitive to $\lambda_0$. This matches our empirical observation that damping becomes most important for larger-scale problems than our Fashion-MNIST setting, and thus has minimal effect here. However, given its substantial importance in these more complex experiments, it is reassuring that the inclusion of damping does not dramatically worsen performance when its influence is not required.

### B.2.4 BATCH SIZE

In Figure 13, we consider each batch size available to ASHA in Section 4.4 ($\{50, 100, 200, 400, 800, 1\,600, 3\,200\}$) to investigate the effect of this hyperparameter on our algorithm.

Since the optimal batch size selected by ASHA for *AdamQLR* was generally large (3 200 in this case), it is perhaps unsurprising that we see divergence from smaller batches. This also matches our intuition: unlike classical first-order methods, *AdamQLR* uses each batch to (implicitly) construct a full curvature matrix for the optimisation surface, which magnifies the importance of having a low-bias sample of the training data. Empirically, we found the computational benefits of fewer batches outweighed the increased cost of computing each batch, so this preference for larger batch sizes aligns with our desire to minimise runtime. Thus, our results show a clear trend that larger batch sizes give greater training and generalisation performance.

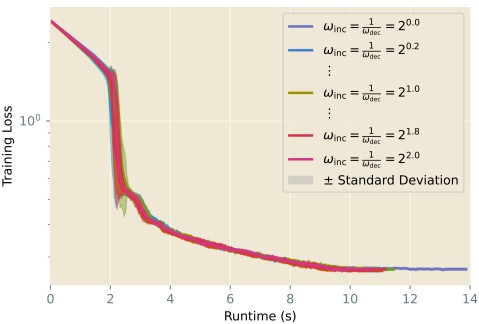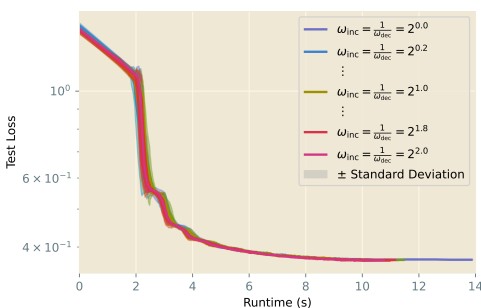

Figure 14: Ablation studies over damping stepping factor for our Fashion-MNIST trial from Section 4.4.

### B.2.5 Damping Stepping Factor

Finally, we explore the effect of different stepping factors by setting $\omega_{\text{inc}}$ to values in $\{2^{0.0}, 2^{0.2}, 2^{0.4}, \cdots, 2^{2.0}\}$, then choosing a symmetric $\omega_{\text{dec}} = \frac{1}{\omega_{\text{inc}}}$. Our results are plotted in Figure 14.

Similarly to learning rate clipping, the impact of different damping stepping factors only becomes most apparent when damping plays a key role in stabilising the optimiser, which does not happen in this Fashion-MNIST test case. However, the plots match our subjective observation that the behaviour at the very start of training is critical to defining the optimisation trajectory, with a high variance at around 2 s of runtime indicating an increased sensitivity here. Moreover, the results reinforce our intuition that the exact factor by which the damping $\lambda$ is modified is not crucially important, so long as AdamQLR is capable of making rapid adjustments over successive optimisation iterations when this becomes necessary.

### B.3 Ablation Studies

In addition to the algorithms plotted in Section 4, we conduct additional experiments to study the impact of different components of *AdamQLR* on its overall performance. Specifically, we examine the effects of Levenberg-Marquardt damping and the choice of curvature matrix used to construct our quadratic model. We use the same experimental configuration as in Section 4, including hyperparameter tuning with ASHA, and plot bootstrapped average trends over 50 repetitions of the best hyperparameters found.

### B.3.1 Levenberg-Marquardt Damping

Appropriate damping is viewed as a necessity in many second-order algorithms in order to defend against degenerate parameter updates, and Figure 15 examines its inclusion in *AdamQLR*. We consider vanilla *Adam* alongside two versions of *AdamQLR*: one which includes damping, and another which excludes it, and perform hyperparameter optimisation as before on each algorithm.

On Fashion-MNIST, we see minimal effect from the inclusion of damping, as the problem does not suffer greatly from degenerate parameter updates. Thus, especially when the internal model of objective space performs well and damping is pushed to very low values, the damping makes a proportionally very small difference to the updates we take. As such, while we do benefit slightly from damping here, the advantage is very slight.

On CIFAR-10, however, we see more dramatic differences from the inclusion of damping, though we note the difference in horizontal scale is likely due to different optimal batch sizes chosen by ASHA. Adjusting for this factor of two, we see very little difference between undamped and damped AdamQLR. This result is surprising — since the model is larger and is substantially more overparameterised than in the Fashion-MNIST case, there are likely to be more parameters to which the output of our network is insensitive, corresponding to low-curvature directions of optimisation space. These low-curvature directions correspond to small eigenvalues of the curvature matrix, so a naïve curvature-based approach would take very large steps in these directions. Because the problem

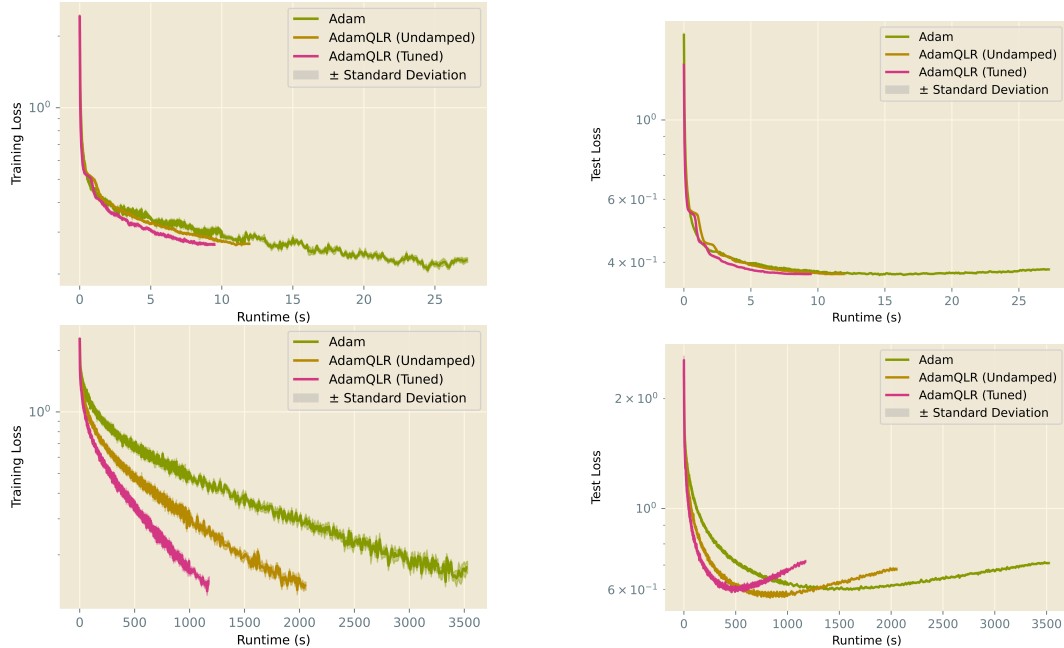

Figure 15: Evolution of Levelberg-Marquardt damping, as measured by Training (left) and Test (right) loss on Fashion-MNIST (top) and CIFAR-10 (bottom)

is inherently non-convex and non-quadratic, such large steps would not be well-motivated, and we would suffer a resulting penalty in our rapidly-excursing loss. However, during our development of AdamQLR, we observed damping to play an important role in avoiding the destabilisation of training. Further, damping clearly stabilises the algorithm enough here to allow for more aggressive optimisation over time; with all this in mind, we retain damping in our default AdamQLR approach.

### B.3.2 CURVATURE MATRIX

As discussed in Appendix C, there is good reason to motivate both the Hessian and the Fisher matrices as curvatures to use to select the learning rate $\alpha$ at each update step. To explore their relative merits, we consider two versions of *AdamQLR*: one which uses Hessian curvature to compute a learning rate and update damping, and another which uses Fisher curvature for the same purposes. The performance of hyperparameter-optimised versions of each setting is compared alongside vanilla *Adam* in Figure 16.

On Fashion-MNIST, we see a slight advantage for Fisher curvature compared to the Hessian curvature, both of which generalise very slightly better than vanilla *Adam*. Curiously, the CIFAR-10 results show the Hessian-based *AdamQLR* technique to make slow progress at the very beginning of training, then proceed similarly to the Fisher version. Again, we note that different optimal batch sizes are likely responsible for most of the horizontal scaling difference. The similarity of these results, combined with the subjectively greater stability of the Fisher version of *AdamQLR* in our development process, justify our use of the Fisher curvature as the default in our algorithm. While Fisher-vector products are more intricate than Hessian-vector products, requiring a rederived component for each loss function, a relatively small number of different loss functions see regular use in practice, so we accept this additional burden.

## C CURVATURE MATRICES: HESSIAN AND FISHER

In this section we discuss in more detail the two main candidates for the curvature matrix $\mathbf{C}$ in our algorithm. Recall from Section 3 that throughout we consider an arbitrary function $f(\boldsymbol{\theta})$ representing the loss function of some network parameterised by $\boldsymbol{\theta}$.

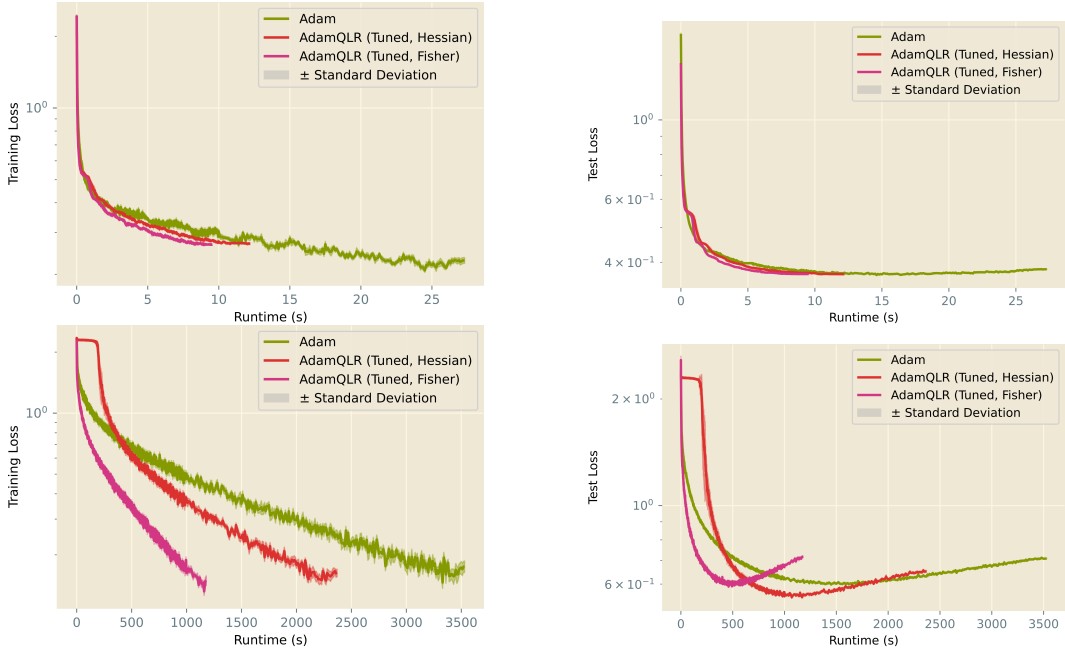

Figure 16: Evaluation of the choice of curvature matrix for the learning rate and damping calculations in *AdamQLR*

## C.1 HESSIAN MATRIX

In this setting, the Hessian curvature matrix follows naturally from the definition of the objective function. A first derivative with respect to $\boldsymbol{\theta}$ yields the gradient vector $\mathbf{g} = (\nabla_{\boldsymbol{\theta}} f)(\boldsymbol{\theta})$, and repeating the derivative yields the Hessian $\mathbf{H} = (\nabla_{\boldsymbol{\theta}} (\nabla_{\boldsymbol{\theta}} f)^{\mathsf{T}})(\boldsymbol{\theta})$.

## C.2 FISHER INFORMATION MATRIX

To draw a connection with the Fisher matrix, we must restate our problem in a probabilistic form. We shall separate the loss function from the neural network, naming the latter $\mathbf{w}_{\boldsymbol{\theta}}(\cdot)$, and consider input-output data pairs $(\mathbf{x}, \mathbf{y})$. Let the input data have some ground truth distribution $p(\mathbf{x})$, and suppose we choose to interpret the output of the network as a probabilistic relationship, such that $\mathbf{w}_{\boldsymbol{\theta}}(\mathbf{x}) = \log p(\mathbf{y}|\mathbf{x})$.

For this model $\mathbf{w}$, the *Fisher Information Matrix* (FIM, or "the Fisher") is defined as:

$$\mathbf{F} = \mathbb{E}_{\mathbf{x} \sim p(\mathbf{x})} \mathbb{E}_{\mathbf{y} \sim p(\mathbf{y}|\mathbf{x})} \left[ \frac{\partial \log p(\mathbf{y}|\mathbf{x})}{\partial \boldsymbol{\theta}} \frac{\partial \log p(\mathbf{y}|\mathbf{x})}{\partial \boldsymbol{\theta}}^{\mathsf{T}} \right]. \quad (4)$$

In its exact form, the Fisher bears many favourable properties for use in optimisation: it is positive semi-definite by construction (so represents a convex space), it is amenable to efficient computation in the form of a matrix=vector product, and provides a parameterisation-independent view of the problem (as in the Natural Gradient Descent (Amari, 1998) family of methods).

Since $\frac{\partial \log p(\mathbf{y}|\mathbf{x})}{\partial \boldsymbol{\theta}}$ is the Jacobian of the network output $\mathbf{w}_{\boldsymbol{\theta}}$ with respect to the parameters $\boldsymbol{\theta}$, the outer product of derivatives is readily available as part of our standard training regime. Although $p(\mathbf{x})$ is unknown, in the mini-batched training setting it is commonly approximated by the empirical distribution $\widehat{p}(\mathbf{x})$ implied by our training dataset. It is important to stress that the expectation of $\mathbf{y}$ is taken with respect to the output distribution of the network, *not* with respect to any ground-truth or empirical distribution $\widehat{p}(\mathbf{y}|\mathbf{x})$ given by the training data. However, some previous work uses the latter distribution as an approximation, resulting in the *empirical* Fisher matrix, which is known to be inferior to the true Fisher.

## C.3 ADAM AND FISHER MATRIX

While Adam is described by its authors as representing an approximation to the Fisher matrix (Kingma & Ba, 2015), we seek here to make the connection more explicit.

The matrix computed inside the expectation of Equation 4 has as its diagonal the elementwise square of $\frac{\partial \log p(\mathbf{y}|\mathbf{x})}{\partial \boldsymbol{\theta}}$. This is connected to the quantity $\mathbf{g}_t = \nabla_{\boldsymbol{\theta}} f(\boldsymbol{\theta}_{t-1})$ computed by Adam; by the chain rule, $\mathbf{g}_t$ is precisely the product of $\frac{\partial \log p(\mathbf{y}|\mathbf{x})}{\partial \boldsymbol{\theta}}$ and the derivative of the loss function with respect to the model output. Neglecting the effect of the latter allows us to view Adam's second-moment buffer $\widehat{\mathbf{v}}_t$ as an approximation to the diagonal of the outer product in Equation 4.

Further, because $\mathbf{g}_t$ is averaged over a mini-batch of input data, we are automatically taking approximate expectations over $\widehat{p}(\mathbf{x})$ and $\widehat{p}(\mathbf{y}|\mathbf{x})$. The approximation arises because the underlying Fisher matrix is not constant, so the contributions from each mini-batch relate to different underlying curvatures. However, the argument motivates the idea that Adam develops an approximation to the diagonal of the empirical Fisher matrix in its buffer $\widehat{\mathbf{v}}_t$.

From this perspective, Adam's elementwise division by the reciprocal of $\widehat{\mathbf{v}}_t$ is simply multiplication by the inverse (approximate) empirical Fisher, and we may interpret $\epsilon$ as a fixed damping term. This picture is slightly corrupted by the square root of $\widehat{\mathbf{v}}_t$ being the quantity actually used by Adam; this operation brings the eigenvalues of the approximate empirical Fisher closer to one, in particular increasing problematic near-zero eigenvalues to more stable values, thus justifying Kingma & Ba's statement that the square root permits more "conservative" preconditioning.

