# OpenReview forum: "Adam through a Second-Order Lens"
_ICLR.cc/2024/Conference — Submitted to ICLR 2024_

### Official Review · Reviewer_2iRB · 2023-10-26

**Soundness:** 2 fair
**Presentation:** 2 fair
**Contribution:** 1 poor
**Rating:** 1
**Confidence:** 5

**Summary:**

This paper proposes AdamQLR, which is a modification of the Adam optimizer and tries to adapt some heuristics used in the K-FAC optimizer for Adam, such as Levenberg Marquardt damping (equation (2) in the paper) and learning rate selection (equation 3 in the paper), both based on a truncated second order Taylor expansion of the function computed by the neural network at the current parameters $\theta_t$ (equation 1 in the paper). The authors perform experiments on 6 tasks (Rosenbrock, UCI Energy/Protein, Fashion-MNIST, SVHN and CIFAR-10) and they compare two versions of their work (Adam QLR Tuned/Untuned) against a few popular optimizers in the literature (SGD Minimal/Full, Adam, K-FAC).

**Strengths:**

Below I enumerate the strengths of the paper:
1. clearly written and easy to understand
2. code is provided
3. optimal hyper-parameters are clearly stated in Table 2
4. ablation study for Levenberg-Marquardt heuristic

**Weaknesses:**

The paper lacks novelty and originality because it only combines Adam and K-FAC in a facile way and thus doesn't provide better results for most of the tasks mentioned, such as UCI Energi/Protein, Fashion-MNIST and CIFAR-10 (in these cases, K-FAC and/or original Adam are better than the proposed method because the generalization performance VS runtime is not competitive as stated in the abstract).

The evaluation was performed on small tasks and I believe that the usage of Rosenbrock function not adding any value to the paper since the tasks that involve Neural Networks are much more complicated. The paper does not have any tables that contains accuracies for classification tasks and it is extremely difficult to figure out what the final accuracies are only by looking at the plots. In the end, it is unfortunate to say that the paper does not meet the novelty and originality requirements for ICLR.

**Questions:**

1. how does AdamQLR behave on NLP tasks?
2. how do you compute the curvature matrix C that is used to update the learning rate $\alpha$ and the damping factor $\lambda$? In the manuscript you state that the overhead is only one additional forward pass, while we all know that computing Hessian-vector-products requires an additional backward pass (which, of course, implies a forward pass in the first place)

---

> ### Author Response · Authors · 2023-11-16
> **Response to Reviewer 2iRB**
>
> Thank you for your efforts reviewing our submission – we appreciate your time.
>
> W1: Since AdamQLR is closely related to its Adam base, we would not expect AdamQLR to display dramatic performance improvements over Adam. The major contribution of our work is that an _Untuned_ version of AdamQLR performs comparably to a _Tuned_ version of Adam. This indicates a significant cost saving which does not appear on our figures: that of performing hyperparameter optimisation. We will update our paper to make this narrative clearer.
>
> While we appreciate our combination of Adam and K-FAC may seem “facile” at first glance, we emphasise our motivating observation that the heuristics used in K-FAC are _essential_ to its ability to perform stable training, rather than providing slight quality improvements. This is a very different relationship from the use of e.g. weight decay in SGD, despite both being ‘heuristics’. From this perspective, we argue an evaluation of these techniques applied to Adam is of interest and value to the ML community, and extends beyond a trivial augmentation of Adam.
>
> W2: Even though our experiments in Figure 2 aren’t as large-scale as some other applications, we think they strike a balance between being large enough to display features of the neural network training problem, but small enough to easily perform multiple repetitions of the experiments to reinforce our results. The Rosenbrock function is included largely as a proof-of-concept, but we think it’s valuable to visualise the trajectory followed by AdamQLR as compared to our benchmarks.
>
> We will add tables of final performance figures to the paper, though we note Figure 4 (Appendix B.1.1) plots accuracy evolutions for our experiments on classification tasks.
>
> To our knowledge, our submission is the first work to apply heuristics from second-order optimisation to a gradient-based method such as Adam. It addresses the relevant question of how these heuristics might aid first-order algorithms, especially considering their fundamental importance to second-order techniques. Further, it describes how the tuned performance of a widely-used optimiser (Adam) can be replicated with AdamQLR without the cost of hyperparameter optimisation. Recalling the ICLR guidance that “a lack of state-of-the-art results does not by itself constitute grounds for rejection”, we feel our work is sufficiently novel and original to be of interest to the ICLR community. We would welcome further discussion to help us understand the root of your disagreement.
>
> Q1: Brief experiments during development suggested our algorithm was a poor match for Penn Treebank trained on the GPT-2 model, so we did not pursue these further. We suspect this may be related to the common observation that transformer models display quite different optimisation dynamics to other settings – indeed, the varying preference for SGD or Adam in particular circumstances has been long-noted in the community. We will add an updated version of this setting to our paper, which we hope will resolve the ambiguity.
>
> Q2: Thank you for pointing out this misstatement. We do indeed compute the curvature matrix $\mathbf{C}$ using the Jacobian-vector product trick, and our implementations may be inspected in the functions `learning_rate_from_hessian()` and `learning_rate_from_fisher()` from `optimisers.py` in the Supplementary Material. These apply a forward-mode Jacobian-vector product to the gradient-producing function, and a forward pass through the gradient is of course a backwards pass through the model. We will update this line accordingly.
>
> We were genuinely surprised by your rating, and would be keen to engage in further discussion to better understand your concerns about our work.

---

> > ### Comment · Reviewer_2iRB · 2023-11-20
> >
> > I would like to give more justification for my rating, point by point with citation from your manuscript and I would appreciate your feedback on each of them.
> >
> > **About computational efficiency of AdamQLR**. I believe that adapting the heuristics of K-FAC to Adam is not a natural approach since Adam is a popular optimizer especially for its simplicity: the covariance matrix of the gradient is used as a proxy for the Hessian matrix, which is supposed to be diagonal, making it easy to compute by squaring the gradient entries, providing a computationally efficient algorithm. I believe that estimating full Hessian information (via the vector products with an additional backward pass) just to compute the learning rate for Adam (that uses the assumption of diagonal Hessian, as described above) just doesn't make sense to me. Moreover, calling this approach computationally-efficient is fundamentally wrong since it incurs  that additional backward pass because, as you have also stated in the paper, Adam is already an adaptive learning rate optimizer.
> >
> >
> >
> > > **We propose a variation of damping based on Adam’s internal curvature estimates which, when applied to Adam’s update proposals, outperforms classical damping from e.g. K-FAC**
> >
> > This statement is unclear to me. How do you measure which damping scheme is better and from which point of view, what is the metric based on which you compare these?
> >
> >
> >
> > > **We might ask if accepting first-order methods’ inaccurate curvature models and applying second-order stability techniques would blend the computational efficiency and optimisation accuracy of each**
> >
> > From your paper I understand that you ran standard K-FAC and I would be interested in whether you tried K-FAC without these heuristics that you inputted to Adam. I believe this should have been a first step in the research flow.
> >
> >
> >
> > **Momentum for K-FAC**. In your paper you skipped the momentum heuristic from K-FAC and state that Adam already has a momentum correction. Indeed, Adam uses momentum for the gradient (similar to how SGD applies it), but K-FAC uses momentum term in such a way that the quadratic approximation M is minimized, which is completely different. Can you please elaborate more on this particular topic, since it is also an heuristic of K-FAC.
> >
> >
> >
> > **Learning rate clipping**. When you rescale the learning rate, you clip it to $\alpha_\text{max}$, which is another heuristic that you introduce. After a quick look at the K-FAC paper, learning rate clipping is not mentioned in the original paper. I believe that blending this heuristic with the ones from K-FAC leads to unfair comparison
> >
> >
> >
> > **Other evaluation flaws**. I believe that it is not fair to compare different optimizers with different batch-sizes, since this parameter yields different number of optimization steps. This also requires manual scaling of initial learning rate to account for the gradient noise in the stochastic gradient.
> >
> >
> >
> > **AdamQLR increases learning rate**. It is known that in the context of stochastic optimization the gradient is noisy, depending on the batch size. The learning rate schedules are designed to decay the learning rate over the course of optimization, converging to zero by the end of training. From SGD convergence analysis we know that by decaying the learning rate we alsodecay the term that depends on the gradient noise which contributes to increasing the upper bound for the convergence (at least in SGD analysis). The simple fact that AdamQLR increases the learning rate might be a problem for why this technique does not yield good results.
> >
> > **Tuning**. It seems to me that your approach needs a lot of tuning in order to make it work, which is an indication that the method is not numerically stable.
> >
> > **Manuscript inconsistencies**. There are some inconsistencies in the information from abstract and introduction which I do not agree with and they are related to all the points that I mentioned in this comment, backed by the observations in the next comment. This is what I meant when I first said that it doesn't meet the ICLR standards.

---

> > ### Comment · Reviewer_2iRB · 2023-11-20
> >
> > I am continuing by pointing out some inconsistencies between the abstract/introduction and the results in Figure 2, which I use to justify my score for the paper.
> >
> > **AdamQLR VS other optimizers**
> > 1. UCI Energy, Figure 2a:
> > - **train loss**: K-FAC < Adam < Tuned AdamQLR < Untuned AdamQLR < SGD (Minimal / Full)
> > - **test loss**: K-FAC has lowest (you could have zoomed in on the interval 0-50 seconds)
> > 2. UCI Protein, Figure 2b:
> > - **train loss**: K-FAC << Tuned AdamQLR < Adam < Untuned AdamQLR < SGD (Minimal / Full)
> > - **test loss**: same relationship as for the train loss
> > 3. Fashion-MNIST, Figure 2c:
> > - **train loss**: K-FAC is by far the best, while AdamQLR (Tuned and Untuned) and original Adam have similar trajectories
> > - **test loss**: K-FAC is the best in the first 5 seconds, then Tuned AdamQLR is better than Adam and Untuned AdamQLR
> > - ** test accuracy**: K-FAC is the best, while Tuned/Untuned AdamQLR and Adam are all similar
> > 4. SVHN, Figure 2d:
> > - **train loss**: here, Untuned AdamQLR is better than all other optimizers in the first ~75s of the training. However, I do not know why there are so many large and frequent decreases in the training loss, compared to the other optimizers, can you please explain that? To me it seems like the learning rate decay is performed more often than for the other optimizers (or is it from the automatic learnign rate adjustment?)
> > - **test loss**: Untuned AdamQLR is the best in the first 50s of the training, while it is almost outperformed by SGD
> > - **test accuracy**: Untuned AdamQLR is similar to SGD
> > 5. CIFAR-10, Figure 2e:
> > - **train loss**: K-FAC < Untuned AdamQLR < Tuned AdamQLR < SGD < Adam
> > - **test loss**: K-FAC decreases the test loss by a lot in the first 250s and is much better than Tuned and Untuned AdamQLR
> > - **test accuracy**: Tuned AdamQLR is better by the time point 1200s
> > 6. ImageNet, Figure 6:
> > - **validation accuracy**: Untuned Adam is better than AdamQLR and SGD
> > - **test accuracy**: same as for the validation accuracy

---

> > > ### Author Response · Authors · 2023-11-22
> > > **Further Response to Reviewer 2iRB**
> > >
> > > Thank you for your very detailed explanation – this has been a great help to understanding your concerns.
> > >
> > > ### Computational Efficiency of AdamQLR
> > > Certainly Adam is a relatively simple algorithm, and this is attractive to developers, but we think its popularity is due also to its practical performance – RMSprop, Adagrad and Lion are comparably simple, but aren’t used nearly as widely. This piques our interest in Adam, as it suggests there may be something about its approach which is particularly amenable to neural network optimisation. It is then valuable to try to understand this behaviour, which can be done by a variety of methods – making Adam more complicated doesn’t obstruct this aim.
> > >
> > > We acknowledge our claims of computational efficiency, which are made from two perspectives, seems confusing from a first-order viewpoint. Firstly: AdamQLR is certainly more efficient than naïve second-order optimisation approaches, since the Hessian-vector product can be computed efficiently and no matrix inversion operations are needed. Secondly: we measure efficiency in abstract terms by the optimisation progress made per unit time, since a method taking slower but more carefully designed steps may be preferable to one which makes rapid, more approximate steps.
> > > Adam does already adapt the learning rate, but our proposal is a more theoretically accurate adaptation heuristic. It then makes sense to compare the quality of the two heuristics.
> > >
> > > ### Damping in AdamQLR
> > > Thank you for pointing out this statement. It is a remnant of a previous draft of our paper, when we were using a different damping mechanism, and no longer applies to this work – we will remove it.
> > >
> > > ### K-FAC Heuristics
> > > We did indeed begin this project with an investigation of K-FAC with its heuristics disabled, and we found that performance was dramatically worsened, with a distinct tendency to diverge during training. As a result, we did not prepare plots for this paper, but this is the foundation for our claim that the heuristics are essential to K-FAC’s success, rather than nice-to-haves.
> > > Even though the mechanics of momentum differ between SGD and K-FAC, the purpose of both remains to interpolate between the update chosen at the current timestep and a decaying memory of previous updates. K-FAC applies momentum by using the previous parameter update to inform the current update direction, then minimises the local model $M$ with both parameters in mind. In this way, K-FAC’s application of momentum is not fundamentally different, but simply incorporates an additional piece of information when looking to minimise the quadratic model. In other words, both methods use momentum to smooth out undesirable gradient dynamics.
> > >
> > > ### Learning Rate Clipping
> > > Learning rate clipping is indeed something we introduce which is not present in K-FAC. We found AdamQLR to be unstable on larger-scale tasks without it, as learning rates could increase in magnitude faster than the adaptive damping heuristic could correct for this. K-FAC has the ability to perform exponential moving averaging over its curvature estimates, which helps ‘smooth out’ regions of problematic curvature, but our use of Jacobian-vector products means we cannot adopt a similar approach. In this sense, we are addressing a behaviour of AdamQLR which K-FAC does not exhibit.
> > >
> > > ### Evaluation on Different Batch Sizes
> > > Our studies found first-order methods tended to prefer small batch sizes, which allow additional noise to be carried through the gradient descent routine, while second-order methods tended to prefer large batch sizes, since these produce more accurate curvature estimates. Fixing a batch size necessarily disadvantages one of these algorithm classes, which we feel would make for an invalid comparison. Furthermore, it would prevent AdamQLR and K-FAC from using larger batch sizes (which would harm generalisation in SGD or Adam) to balance their higher cost per step. Naturally, this means every optimiser uses different numbers of steps, but we compare according to runtime to avoid this becoming a problem. There is unfortunately not a perfect way of performing this comparison.
> > >
> > > While we acknowledge different batch sizes may require learning rate rescaling, we believe this is covered by our hyperparameter optimisation strategy, which is able to select learning rates appropriate for the setting.
> > >
> > > ### AdamQLR Learning Rates
> > > We agree the decaying learning rate is an important feature of the convergence of SGD, and an analysis of the gradient (and curvature) noise affecting AdamQLR would be interesting. However, we note our learning rates in Figure 5 never rise to unrealistic levels, and it is a common feature of second-order methods that learning rates are chosen based on model curvature without any explicit decay behaviour.
> > >
> > > _(continued)_

---

> > > > ### Author Response · Authors · 2023-11-22
> > > > **Further Response to Reviewer 2iRB (continued)**
> > > >
> > > > ### Hyperparameter Tuning
> > > > We are unsure how you conclude that our method requires a large amount of hyperparameter tuning. It receives exactly the same hyperparameter optimisation treatment as the other benchmarks, and that treatment is reasonably modest in scale and cost. Further, we specifically show results on _AdamQLR (Untuned)_ to emphasise that these default hyperparameters lead to robust behaviour – we emphasise again that the performance of this untuned baseline is a major thrust of our contribution.
> > > >
> > > > ### Inconsistencies
> > > > In our revised abstract, we claim “an _untuned_ AdamQLR setting achieves comparable generalisation performance vs runtime to _tuned_ benchmarks”. Our revised introduction claims “the result is an efficient, scalable algorithm whose untuned form competes strongly with commonly-used optimisers, demonstrating robustness to its few remaining hyperparameters”, and “our untuned method competes with methods using tuned hyperparameters”.
> > > >
> > > > By these remarks, we mean to say that AdamQLR – especially its untuned version – performs comparably to methods such as SGD and Adam which have undergone hyperparameter tuning. Achieving similar performance to baselines while saving the cost of hyperparameter tuning is a substantial benefit, so it should not matter that we do not ‘beat’ every baseline. Perhaps our algorithm’s performance would be more clearly described if we replace “compete” with “comparable” in the above quotes, which we can do if desired, but we feel all those claims fairly reflect our results, in which AdamQLR exhibits behaviour in line with widely-used optimisers.

---

### Official Review · Reviewer_MsHa · 2023-11-01

**Soundness:** 3 good
**Presentation:** 3 good
**Contribution:** 3 good
**Rating:** 6
**Confidence:** 3

**Summary:**

This work refers to Adam and proposes to adaptively adjust the learning rate. Specifically, the authors utilize $\rho$ to denote the ratio between the difference of true loss function $f()$ and the difference of second-order estimation $M()$. Then the authors refine the estimated Hessian matrix through $\lambda$ according to $\rho$. Finally, the learning rate is then computed by minimizing $M(\theta - \alpha d)$.

**Strengths:**

1.	The method makes sense.

2.	Extensive experiments show the effectiveness of the method.

**Weaknesses:**

1.	I wonder how to get the matrix $C$ in Eq. 1.

2.	What is the principle of setting $\omega_{dec}$ and $\omega_{inc}$, and why $\lambda$ is adjusted when $\rho$ larger than 3/4 or smaller than 1/4?

**Questions:**

Please see the weaknesses.

---

> ### Author Response · Authors · 2023-11-16
> **Response to Reviewer MsHa**
>
> Thank you for your efforts reviewing our submission – we appreciate your time.
>
> W1: The curvature matrix $\mathbf{C}$ denotes any estimate of objective curvature we may wish to use. AdamQLR only uses this matrix in the context of a product with an arbitrary vector. Using the Jacobian-vector product trick (Pearlmutter, 1994; reference in paper), we can compute such products with the Fisher information matrix (which we use) or the Hessian matrix (a common alternative) without having to compute or store the whole matrix. This allows us to compute the exact matrix-vector products according to the definition of our chosen curvature matrix. The functions `learning_rate_from_hessian()` and `learning_rate_from_fisher()` in `optimisers.py` from our Supplementary Material contain our implementations in JAX. We briefly mention this approach in the final paragraph of Section 3.3, and will expand our explanation to make this clearer.
>
> W2: In principle, we seek $\omega_\textrm{dec}$ and $\omega_\textrm{inc}$  to be sufficiently close to 1 to avoid destabilising the optimiser with dramatically-varying damping, but sufficiently far from 1 to allow prompt rectification of any undesired behaviour from the optimiser. We note that a major thrust of our contribution is that an _Untuned_ version of AdamQLR (using the default hyperparameters we suggest) performs comparably to a _Tuned_ version of Adam, which avoids placing any burden of selecting these factors on the end-user – we will update our paper to make this clearer. In any case, our sensitivity study in Figure 13 (Appendix B.2.5) suggests AdamQLR is robust to the setting of these factors.
>
> Our thresholds for $\rho$ in Equation 2 were taken directly from the original K-FAC settings (Martens and Grosse, 2015; reference in paper), since we considered them to be components of the damping strategy we were porting to Adam. Their purpose is to detect when the loss change predicted by the model is much larger than the actual change, and intervene quickly to apply damping to mitigate the model error. The intention is to bias the decision towards decreasing damping if the optimisation is proceeding stably, since in the vicinity of a local minimum excessive damping would slow down convergence. We will add a more verbose discussion of this phenomenon to the paper.

---

### Official Review · Reviewer_mFeq · 2023-11-01

**Soundness:** 1 poor
**Presentation:** 3 good
**Contribution:** 1 poor
**Rating:** 3
**Confidence:** 4

**Summary:**

In this paper authors propose some symbiosis of two optimization methods: Adam and K-FAC. They combine damping and learning rate selection techniques from K-FAC and use it inside Adam algorithm. The resulting algorithm, called AdamQLR, is then evaluated on different regression and classification tasks.

**Strengths:**

1. Lots of numerical experiments.
2. Clear description of algorithm modification.
3. Good description of the motivation of the heuristics, adopted from K-FAC.
4. Description of the experimental setup and hyperparameter search space.

**Weaknesses:**

From theoretical point of view, the result seems insignificant. You took some heuristics, that improve the model, and moved it to another model. There is no evidence, that it should work better in theory. From practical point of view, as far as I understand, the number of hyperparameters increased: $\beta_1, \beta_2, \varepsilon$ for Adam vs $\beta_1, \beta_2, \varepsilon, \lambda$ for AdamQLR (or even $w_{dec}, w_{inc}$ instead of $\lambda$.

**Questions:**

Rosenbrock function example seems unfair, because you use Hessian there, what do you think?

---

> ### Author Response · Authors · 2023-11-16
> **Response to Reviewer mFeq**
>
> Thank you for your efforts reviewing our submission – we appreciate your time.
>
> W: We agree our work has a greater empirical focus than theoretical. We believe valuable contributions can be made from both perspectives, and the clear intuition behind our method partly substitutes a more theoretically robust (but less practically useful) analysis.
>
> While the features we port from K-FAC are heuristics, we feel this slightly undersells their importance, because in our experience these heuristics are _essential_ to K-FAC making progress in training without diverging, rather than being nice-to-haves which incrementally polish the performance of an already-strong underlying algorithm. A main theme of our work is thus to investigate the possibility that these heuristics can play a more fundamental role in optimisation. We will revise our paper to make this point more clearly.
>
> We do indeed introduce additional damping hyperparameters for AdamQLR. However, a major thrust of our contribution is that the _Untuned_ version of AdamQLR performs comparably to the _Tuned_ version of Adam, so we claim it is reasonable to leave these hyperparameters at their default values in practice. Indeed, the fact that damping is adaptive means AdamQLR can effectively correct its own hyperparameters to an extent. We will update our paper to place a greater emphasis on this contribution.
>
> Q: Unlike our neural network tasks, the Rosenbrock function does not output a probability distribution, so we cannot compute a Fisher information matrix. AdamQLR supports any curvature estimate, so we use the Hessian matrix as the next-best option. We note that Hessian-vector products allow multiplication by large Hessians at similar cost to the Fisher-vector products we use in the other training tasks, so there is not a computational disparity between the Rosenbrock experiment and our other evaluations. If by “unfair” you mean a different issue, could you please clarify?

---

> ### Comment · Reviewer_mFeq · 2023-11-21
>
> Thanks for your comments and changes!
> I have been looking through the revised version more carefully, and I've got several remarks.
> 1. You say, that your main motivation is to show that untuned version of AdamQLR performs similarly to tuned Adam. However you do not provide the results for untuned Adam. Maybe it performs similarly.
> 2. Again, you do not provide the results for untuned K-FAC. Thus, we do not know, how it performs on these tasks.
> 3. FashionMNIST: you say, that K-FAC overfits much earlier, compared to other methods. But It achieves the best test accuracy faster, then any other method. So it is a wrong conclusion: if we stop all the methods earlier, and not when K-FAC starts overfitting, it will be the best.
> 4. Actually, K-FAC performs the best in most of the experiments. And, if you say, that your main motivation is not to provide SOTA method, but to provide a method, which untuned version has comparable performance with tuned Adam or K-FAC, then again see points 1 and 2.
> 4. The section about batch size seems weird, since obviously the bigger batch size, the better convergence of the algorithm, since it narrows the area of convergence of stochastic gradient-based method, which can be seen from convergence rate of SGD [1].
> 4. You only provide performance results against time, which seems not enough, and it is better to provide also performance against epochs.
> 4. When we look at experiments on bigger models and datasets (ImageNet, Penn Treebank), we see, that proposed method is outperformed by all the others. Taking into account my points 1-4, it seems unfair to say, that untuned AdamQLR performs on the same level as tuned Adam on any task.
>
> [1] Gower, Robert Mansel, et al. "SGD: General analysis and improved rates." International conference on machine learning. PMLR, 2019.

---

> > ### Author Response · Authors · 2023-11-23
> > **Further Response to Reviewer mFeq**
> >
> > Thank you for your further analysis of our paper, and for listing more action points for a future version. While time is too short to add a new revision in the discussion period, we would note:
> >
> > 1, 2, 4: We agree results on untuned optimisers would be a natural fit for our work. However, since the community very broadly uses tuned versions of SGD and Adam, it seems almost certain that these would be significantly more useful than untuned default variants.
> >
> > 3: Indeed, including early stopping would alter our results. We have not used it to simplify our evaluation, but would note that in many practical circumstances, a robustness to overfitting is just as important as being able to transiently achieve stronger generalisation.
> >
> > 5: In the context of AdamQLR, our discussion of batch size is less to focus on the convergence properties of the algorithm, and more to study the accuracy of the curvature estimates produced by each batch. These have an immediate impact on the stability and robustness of second-order algorithms, with divergence potentially resulting from poor choices, so this is our greater priority.
> >
> > 6: We can certainly extend our results to include epoch-wise plots. Our concern is that these would impart an unfair advantage to methods incorporating more second-order principles, since they would not be fairly penalised for the greater computational complexity of each step. Ultimately, we believe runtime provides a fairer comparison which more reflects the interests of the practitioner.

---

### Official Review · Reviewer_BW7Y · 2023-11-07

**Soundness:** 3 good
**Presentation:** 3 good
**Contribution:** 2 fair
**Rating:** 6
**Confidence:** 3

**Summary:**

This paper tried to combine the first-order method (such as Adam) with the second-order methods, such as K-FAC. More specifically, the authors propose a novel optimizer AdamQLR: combining damping and learning rate selection techniques of K-FAC. The experimental results illustrate that the proposed method AdamQLR can achieve competitive generalisation performance and training efficiency.

**Strengths:**

1. The idea of combining first-order and second-order methods is very interesting. In addition, the research direction is also very important.
2. The proposed method is very easy to understand. I think we should pay more attention to second-order method and improve its efficiency.

**Weaknesses:**

1. I think the main results are from the figure 2. But the figure is not very clear for me, maybe you can list the training loss, test loss, convergence steps, and generalization gap ( |training_loss - test_loss| ) in a table. From this figure. I'm not very clear whether the proposed method can solve the overfitting issue and improve the generalization. So I think you can analyze the generalization gap.

2. The experimental results are not very strong for me. Although the proposed method can achieve fast convergence and lower test loss, their performance is still too close. In addition, you try to analyze training loss and test loss in figure 2. But loss value is not a great metric for classification tasks and I think you should show the accuracy.

3. The training task is too simple and the results on complex tasks (such as ImageNet) is not very strong.

**Questions:**

1. Loss value in figure 2 is not great enough to compare the performance of different methods and maybe you should provide the accuracy value.

---

> ### Author Response · Authors · 2023-11-16
> **Response to Reviewer BW7Y**
>
> Thank you for your efforts reviewing our submission – we appreciate your time.
>
> W1/Q1: Figure 2 is indeed the crux of our results section, and our claims about generalisability are based on comparisons from these graphs. We will add a table of final results as you suggest.
>
> W2: We show classification accuracies in Figure 4 (Appendix B.1.1), since we agree there is reason to be interested in both loss and accuracy performance. We will update the main body to point out this figure more clearly. Regarding the performance similarities: since we apply second-order heuristics to the core Adam algorithm, we don’t expect to massively beat it – our main contribution is that the _Untuned_ version of AdamQLR performs comparably to the _Tuned_ version of Adam, meaning AdamQLR (Untuned) can be applied with dramatically reduced computational effort.
>
> W3: Even though the tasks in Figure 2 aren’t as large-scale as some other applications, we think they strike a balance between being large enough to display features of the neural network training problem, but small enough to easily perform multiple repetitions of the experiments to reinforce our results. The reviewer may also be interested in the Penn Treebank/GPT-2 results we will be adding in response to other feedback.

---

### Author Response · Authors · 2023-11-17
**New Paper Revision following Reviewer Feedback**

Thank you to all the reviewers for your time and comments – they are invaluable to our efforts to improve our work.

We have now uploaded a revision based on your suggestions, with the following main changes:
* We include tables of numerical results for our experiments of Section 4 in Table 5 (Appendix B.1.5), with the fixed-runtime comparisons in Appendix B.1.6 also receiving numerical results in Tables 6 and 7.
* We now include experiments training a GPT-2 model on Penn Treebank (Appendix B.1.4 - Figure 7 and Table 5).
* A new note in Appendix A.4 discusses our damping adaptation in Equation 2 in more detail, giving intuition and justification for the values used.
* We have made minor edits to the narrative throughout to emphasise our key contribution: that an _Untuned_ version of AdamQLR performs comparably to _Tuned_ versions of popular benchmarks.
* We have edited our experimental commentary to emphasise that results on accuracy metrics are available in Figure 4 and Table 5 (Appendix B.1.1 and Appendix B.1.5), and results under a fixed-runtime hyperparameter tuning methodology are available in Figures 8 and 9 and Tables 6 and 7 (all in Appendix B.1.6).
* We have edited our discussion of K-FAC heuristics to emphasise they are _essential_ to the algorithm's function, rather than 'nice-to-haves' like weight decay for SGD.
* We have clarified our discussion in Section 3.4 of how the curvature matrix $\mathbf{C}$ is calculated, and corrected our statement of the cost of computing Jacobian-vector products in the same Section.

In the interests of making new results available ASAP, we have made these changes to the original submission, so this revision is around one third of a page too long. We are preparing additional edits to restore the correct paper length, and will share these as soon as they are available.

We would be very grateful to hear of any further feedback on our paper, including elements you would like to see changed to strengthen the work.

---

> ### Comment · Reviewer_mFeq · 2023-11-21
> **Highlights of changes**
>
> Please, if it is possible highlight all the changes that you made in the original paper.

---

> > ### Author Response · Authors · 2023-11-22
> > **Complete Changelog**
> >
> > No problem - here are all the individual changes, based on our Overleaf changelog (new text is all in quotation marks):
> > * Abstract
> >   * Added that second-order methods "only function effectively with the addition of" stabilising heuristics
> >   * Added "finding an _untuned_ AdamQLR setting" achieves "comparable" generalisation performance vs runtime "to _tuned_ benchmarks"
> > * Section 1: Introduction
> >   * Italicise "depend" in paragraph 2, sentence 2
> >   * Added entire sentence: "Heuristics commonly seen in first-order methods, such as weight decay or momentum applied to SGD, improve an already effective optimiser; by contrast, second-order methods' heuristics are _essential_ components, without which the optimiser will perform unstably or ineffectively"
> >   * Paragraph 3, sentence 2, add scalable algorithm "whose untuned form" competes strongly with "tuned" commonly-used optimisers, "demonstrating robustness"
> >   * Final bullet point: our "untuned" method competes with methods using tuned "hyperparameters", exhibiting robustness to hyperparameters "while saving substantial tuning cost"
> > * Section 2: Related Work
> >   * Add citation "Niu et al. (2023) uses a parallel approach to ours to incorporate momentum into L-BFGS (Liu & Nocedal, 1989)"
> > * Section 3: AdamQLR
> >   * Section 3.1, paragraph 3, last sentence: emphasis on "essential"
> >   * Algorithm 2: add words "learning rate" and "damping" in pink text
> >   * Section 3.3, paragraph 1: added: includes three "important" stabilising heuristics
> >   * Section 3.3, after (2), add "We discuss this formulation further in Appendix A.4."
> >   * Section 3.3, final paragraph: fix: compute these quantities using only one additional "backwards" pass per product with $\mathbf{C}$
> >   * Section 3.4, paragraph 2, add last sentence "Our implementation exploits Jacobian-vector products and the efficient Fisher decomposition described in Martens & Grosse (2015, Appendix C), which computes exact products without explicitly storing $\mathbf{C}$."
> >   * Section 3.4, paragraph 3, last sentence, add ", and that larger batch sizes tended to improve our curvature estimates, leading to better performance despite the higher cost of each forward pass."
> >   * Section 3.4, paragraph 4, add last sentence "We justify this claim in Section 4."
> >   * Section 3.4, last paragraph, edit first sentence: we suffer additional forward "and backward" passes
> > * Section 4: Experiments
> >   * After list of tasks, extend first sentence: ", and a study on Penn Treebank in Appendix B.1.4."
> >   * Paragraph after list of algorithms, add last sentence "With the exception of the Rosenbrock Function, we give a numerical comparison of the end-of-training statistics in Table 5."
> >   * Before Section 4.1, add new paragraph "In Appendix B.1.6, we present analogous results where the hyperparameters are tuned to minimise training or validation losses after a fixed runtime, without constraining the number of epochs."
> >   * Figure 2: extend caption: ", and numerical comparison in Table 5."
> >   * Section 4.2, paragraph 3: edit: Generally, _AdamQLR (Tuned)_ "and _Untuned_" compete comparably with...
> >   * Section 4.3, paragraph 2, edit sentence 2: still see meaningful benefit from the "_AdamQLR_" algorithm, "with the _Tuned_ variant" now comfortably outperforming _Adam_
> >   * Section 4.4, paragraph 1, extend last sentence: ", and accuracy evolutions in Figure 4a."
> >   * Section 4.5, paragraph 1, extend sentence 1: Figures 2d "(losses) and 4b (accuracies)."
> >   * Section 4.5, paragraph 3, last sentence, change first word to "Additionally"
> >   * Section 4.6, paragraph 1, extend last sentence: "and accuracy results in Figure 4c."
> > * Appendix A.4: Reduction Ratio - all new
> > * Appendix B.1.1, Figure 4 and Figure 5: add references to numerical comparisons in Table 5
> > * Appendix B.1.3: ImageNet
> >   * Paragraph 1, extend ", using untuned _Adam_ and _AdamQLR_ baselines alongside their ‘SGD + Heavy ball momentum’ setting (which we we call _SGD-ImageNet_).
> >   * Figure 6: now showing training and test accuracy, with validation cut
> > * Appendix B.1.4: Penn Treebank and Figure 7 - all new
> > * Appendix B.1.5: Numerical Results and Table 5 - all new
> > * Appendix B.1.6: Fixed-Runtime Comparisons
> >   * Add Tables 6 and 7, and references
> >   * Paragraph 2, sentence 1, add: later phases of training "on Fashion-MNIST"
> >   * Paragraph 2, sentence 2, change last phrase: behaviour on "larger datasets".
> >   * Paragraph 2, insert sentence 3: "However, on CIFAR-10 _AdamQLR (Tuned)_ achieves the strongest generalisation, and even on SVHN its performance is competitive."
> >   * Paragraph 2, last sentence, extend ",  larger datasets achieve particularly strong training loss evolutions."
> >   * Figures 8d, 8e, 9d and 9e all new

---

> > > ### Comment · Reviewer_mFeq · 2023-11-22
> > >
> > > Thanks, but actually I meant highlight with color in the revised PDF.

---

> > > > ### Author Response · Authors · 2023-11-22
> > > > **New Marked-Up Difference PDF Available**
> > > >
> > > > Our apologies for the misunderstanding. We have now updated our Supplementary Material to include the output of `latexdiff` applied to our original and revised submissions, which is hopefully what you had in mind.

---

### Meta-Review · Area_Chair_LHHr · 2023-12-08

**Metareview:**

- The submission explores integrating first-order methods like Adam with second-order techniques, specifically K-FAC. The authors introduce a new optimizer, AdamQLR, which merges K-FAC's damping and learning rate optimization strategies. Experimental outcomes demonstrate that AdamQLR attains comparable results in terms of generalization and training efficiency.
- However, this submission still has some shortcomings:
  - The experiment is not convincing. The scale for the numerical result is quite small and the setting is kind of native. It is hard to evaluate its performance in the advanced models training.
  - For training DNN models, this ML community has its own (but maybe heuristic way) to adjust the LR in each iteration, like warmup+cosin decay. This submission should discuss the advantages of choosing AdamQLR instead of the popular heuristic ways.
  - AdamQLR lacks computational efficiency.  As pointed out by the Reviewer, AdamQLR is still less efficient and provides worse results than Adam in some cases.
  - From the added experimental results, AdamQLR does not seem to easily surpass the baselines.

**Justification For Why Not Higher Score:**

- The submission contains too many shortcomings which is hard to fix with minor revision.
- As pointed out by the reviewers, some comparisons in the experimental parts lack fairness.

**Justification For Why Not Lower Score:**

N/A

---

### Decision · Program_Chairs · 2024-01-16

Reject